# Leukemia relapse via genetic immune escape after allogeneic hematopoietic cell transplantation

Simona Pagliuca [1,2,3,13], Carmelo Gurnari [1,4,13], Colin Hercus[5], Sébastien Hergalant [6], Sanghee Hong[7], Adele Dhuyser[3,8], Maud D'Aveni[2,3], Alice Aarnink[3,8], Marie Thérèse Rubio[2,3], Pierre Feugier[2], Francesca Ferraro [9], Hetty E. Carraway [10], Ronald Sobecks[11], Betty K. Hamilton [11], Navneet S. Majhail[12], Valeria Visconte [1] & Jaroslaw P. Maciejewski [1] ✉

Graft-versus-leukemia (GvL) reactions are responsible for the effectiveness of allogeneic hematopoietic cell transplantation as a treatment modality for myeloid neoplasia, whereby donor T- effector cells recognize leukemia neoantigens. However, a substantial fraction of patients experiences relapses because of the failure of the immunological responses to control leukemic outgrowth. Here, through a broad immunogenetic study, we demonstrate that germline and somatic reduction of human leucocyte antigen (HLA) heterogeneity enhances the risk of leukemic recurrence. We show that preexistent germline-encoded low evolutionary divergence of class II HLA genotypes constitutes an independent factor associated with disease relapse and that acquisition of clonal somatic defects in HLA alleles may lead to escape from GvL control. Both class I and II HLA genes are targeted by somatic mutations as clonal selection factors potentially impairing cellular immune responses and response to immunomodulatory strategies. These findings define key molecular modes of post-transplant leukemia escape contributing to relapse.

Hematopoietic cell transplantation (allo-HCT) holds a pivotal place in the therapeutic algorithm of myeloid neoplasms (MN)[1,2]. Either as an upfront treatment in diseases with low blast burden, after disease control in myelodysplastic syndromes (MDS) or following consolidation therapy in intermediate and high-risk acute myeloid leukemia (AML), its therapeutic index is principally related to the immunogenic graft-versus-leukemia (GvL) effect, exerted by donor-derived immune effectors on leukemic cells[3]. Relapse following allo-HCT may be mechanistically multifactorial and includes re-expansion of the residual leukemia with or without acquisition of novel driver lesions along with the development of various immune escape mechanisms contributing to the evasion from GvL effect[4,5].

[1]Department of Translational Hematology and Oncology Research, Taussig Cancer Institute, Cleveland Clinic, Cleveland, OH, USA. [2]Department of Hematology, CHRU de Nancy, Vandœuvre-lès-Nancy, France. [3]CNRS UMR 7365, IMoPA, Biopole of University of Lorraine, Vandœuvre-lès-Nancy, France. [4]Department of Biomedicine and Prevention, PhD in Immunology, Molecular Medicine and Applied Biotechnology, University of Rome Tor Vergata, Rome, Italy. [5]Novocraft Technologies Sdn Bhd, Kuala Lumpur, Malaysia. [6]Inserm UMR-S 1256 Nutrition-Genetics-Environmental Risk Exposure, University of Lorraine, 54500 Vandœuvre-lès-Nancy, France. [7]Division of Hematologic Malignancies and Cellular Therapy, Department of Medicine, Duke University School of Medicine, Durham, NC, USA. [8]Histocompatibility Department, CHRU de Nancy, Vandœuvre-lès-Nancy, France. [9]Division of Oncology, Department of Medicine, Washington University School of Medicine in St. Louis, St. Louis, MO, USA. [10]Leukemia Program, Hematology Department, Taussig Cancer Institute, Cleveland Clinic, Cleveland, OH, USA. [11]Blood and Marrow Transplant Program, Taussig Cancer Institute, Cleveland Clinic, Cleveland, OH, USA. [12]Sarah Cannon Transplant and Cellular Therapy Network, Nashville, TN, USA. [13]These authors contributed equally: Simona Pagliuca, Carmelo Gurnari. ✉e-mail: maciejj@ccf.org

Post-transplant alloreactivity ideally acts via class I and II human leukocyte antigen (HLA)-restricted mechanisms. Therefore, one of the biological modes of relapse after allo-HCT is the loss of expression of disparate immunodominant HLA alleles via loss of heterozygosity (LOH) or epigenetic down-regulation, which have been demonstrated in haploidentical, mismatched unrelated and matched related settings[6-8]. While large genomic aberrations including the loss or the duplication of an entire haplotype or of a single allele have been noted across various studies[6,9-11], more granular lesions of the HLA region including microdeletions and point mutations, remain underestimated in allo-HCT. Such lesions, ultimately resulting in phenotypes of immune resistance, have been reported in the context of clonal adaptive recovery in immune-mediated aplastic anemia (AA) or as immune escape drivers in solid tumors, together with up-modulation of negative immune checkpoint regulators[12-14].

HLA heterogeneity and the type of HLA polymorphisms in certain loci have also been found as factors influencing the direction of alloreactive responses[15-19]. HLA evolutionary divergence (HED), a metric reflecting the immunopeptidome diversity, has been recently linked to post-immunotherapy outcomes in solid tumors[20]. Furthermore, we and others demonstrated that recipient HED correlates with the efficiency of immune reconstitution after allo-HCT[21,22]. Since this score mirrors the antigenic spectrum capacity, it may be considered a surrogate of the individuals' ability to present leukemia associated antigens (LAA) in transplant recipients[20,23,24].

In analogy with solid cancer immunotherapy, we hypothesized that, also in the context of allo-HCT, a reduced variability of HLA genotypes as well as genomic mechanisms of escape can converge in overlapping immune-escape phenotypes.

Here we investigate relapsed leukemia in allo-HCT recipients for the presence of non-driving genomic lesions in immune elements as part of the mutational landscape of recurrent leukemia. To unravel the genomic dysfunctions leading to immune-escape, we study HED and both genetic and transcriptomic inactivation of HLA via somatic mutations or down-modulation as determinants of leukemic relapse in the setting of allo-HCT.

We demonstrate how both germline and somatic dysfunction of HLA heterogeneity provides an immunogenetic framework prone to post-transplant leukemia immune escape and relapse.

## Results

### Dysfunction of germline HLA heterogeneity as immunogenetic driver of post allo-HCT relapse

We started from the concept that a more structurally diverse HLA allele combination could widen the spectrum of antigen presentation, thereby enhancing the GvL effect. We thus analyzed a cohort of 494 patients (median follow up 27.5 [14–45] months) who received allo-HCT for MN and had available high-resolution HLA genotype (Table 1; Fig. 1; Data Source 1). For each patient we calculated per-locus, per-class and global HED scores. This parameter quantitates the physio-chemical divergence existing between two amino acids within the peptide binding cleft of two homologous HLA alleles. To investigate the impact of HED distribution on clinical outcomes, we categorized each value using a cutoff that could be standardized and clinically meaningful. Since specific polymorphisms in different HLA loci have been previously associated with AML susceptibility or relapse[25-27] thereby potentially biasing global HED scores, we did not base our study on cutoff values computed on this patient population. Instead, for the assessment of this metric we used cutoffs defined in healthy controls (HC) (Supplementary Fig. 1; Supplementary Data 1) and considered as a benchmark the 50th percentile of these HC HED scores (locus-specific and global), as we previously described[28].

In the matched allo-HCT context (N = 393), we found that patients with high class II HED had a lower relapse rate as compared to those with low class II HED values (hazard ratio [HR]: 0.65, 95% CI: 0.45 0.958,

### Table 1 | Patient and transplant characteristics of the whole cohort (N = 494)

|  |  | N (% or IQR) |
|---|---|---|
| Whole cohort |  | N = 494 |
| Median Age at HCT (years) |  | 59 (52–65) |
| Disease |  |  |
|  | AML | 294 (59) |
|  | MDS | 125 (26) |
|  | MPN | 75 (15) |
| Donor type |  |  |
|  | MRD | 130 (26) |
|  | MUD | 261 (53) |
|  | MMUD | 6 (<1) |
|  | Haplo | 97 (20) |
| Graft type |  |  |
|  | BM | 153(31) |
|  | PBSC | 341 (69) |
| Conditioning regimen |  |  |
|  | MAC | 176 (36) |
|  | RIC | 318 (64) |
| Post-transplant relapse |  |  |
|  | Yes | 142 (29) |
|  | No | 352 (71) |
| Median time to relapse (days) |  | 173 (100–319) |
| Median follow-up (months) |  | 27.5 (14–45.3) |

IQR interquartile range, HCT hematopoietic cell transplantation, AML acute myeloid leukemia, MDS myelodysplastic syndrome, MPN myeloproliferative neoplasms, MRD matched related donor, MUD matched unrelated donor, MMUD mismatched unrelated donor, Haplo haploidentical donor, BM bone marrow, PBSC peripheral blood stem cells, MAC myeloablative, RIC reduced intensity.

$p = 0.021$). A breakdown of the specific contribution of the class II loci, showed that HLA-DQB1 and HLA-DPB1 impacted the most on this result ($p = 0.04$, Fig. 2a, b). Among class I loci, the only impact was seen for HLA-C, for which high HED values were associated with lower cumulative incidence (CI) of relapse ($p = 0.041$). Class II, but not class I HED, affected the probability of relapse also when considered in cause-specific multivariable regression models (HR: 0.54, 95% CI: 0.36–0.81, $p = 0.003$), adjusted for age, type of donor, conditioning regimen, disease type, stem cell source, HCT-CI score, disease risk and year of transplant (Fig. 2c, d). In a similar fashion, high class II HED values were associated with better overall survival (OS HR: 0.63, 95% CI: 0.46–0.86, $p = 0.004$), whereas no correlation was observed with acute and chronic GvHD (Supplementary Fig. 2).

For the haploidentical and mismatched unrelated population (N = 100), neither class I, nor class II or global HED were found to affect disease recurrence (Supplementary Fig. 3), implying that a modulation of the immunogenetic risk by other transplant-related factors may influence the GvL effect.

To link immune control dysfunction with abnormal HLA heterogeneity, we performed deep targeted next generation sequencing (NGS) for T cell receptor (TCR) Vβ complementarity determining region 3 (CDR3) after 3–6 months following allo-HCT (N = 24) and characterized the specificity of TCR repertoire using previously described bioinformatic frameworks[21]. Overall, patients with higher class I and II HED were characterized by a less expanded repertoire together with increased cancer-related clonotypes and less clonally-polarized features compared to those with lower divergence. This phenomenon was also recorded for pathogen-recognizing clonotypes, underlying a general imprinting for a more diverse and variegated antigenic response (Supplementary Fig. 4). We asked whether the differences in repertoire expansion could be related to a higher frequency

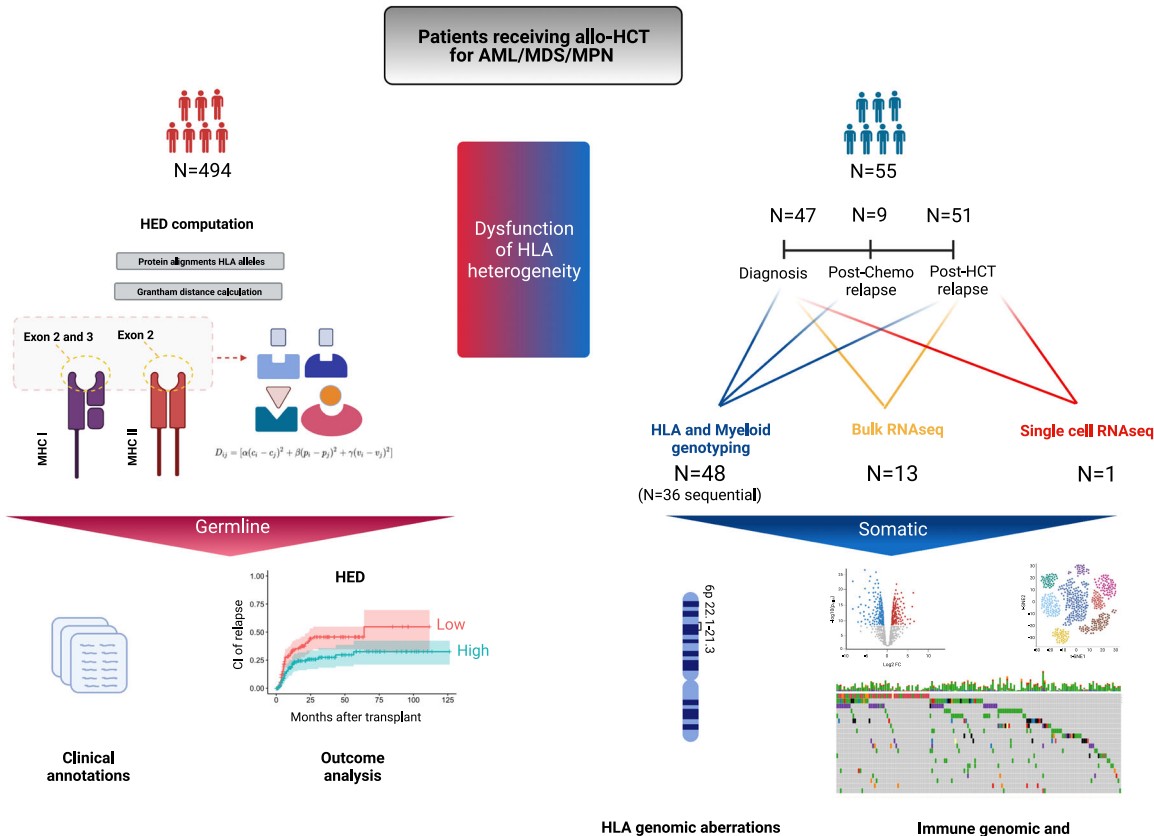

**Fig. 1 | Study design.** In total 494 patients receiving a first allogeneic hematopoietic cell transplantation (HCT) for a myeloid disorder were included in this study. For the estimation of the dysfunction of the germline HLA diversity, we calculated the HLA evolutionary divergence, a metric able to quantitate the structural diversity at the peptide binding site of each homologous HLA molecule (LEFT panel). This metric was correlated in univariable and multivariate analysis with the cumulative incidence of relapse and other clinical outcomes. To characterize the somatic dysfunction of HLA diversity (RIGHT panel) we performed a longitudinal immunogenomic and transcriptomic analysis in a subgroup of patients relapsing after HCT (N = 55). Classical HLA loci and myeloid genes were genotyped, and their mutational status was analyzed. In addition, bulk and single cell RNA sequencing were also performed. Of note is that 7 samples used for the bulk RNA sequencing were previously published (Christopher et al. NEJM 2018).

of CMV reactivation in relation to HED values; but no difference in terms of CMV infection was observed between patients with low vs high class I and II HED ($p = 0.07$ for class I and $p = 0.691$ for class II, Supplementary Data 2). Longitudinal clonotype tracking analysis showed that some of these clones were present in pre-transplant or donor samples predating post-transplant hyper-expansion (Supplementary Fig. 5).

## HLA and myeloid landscape

We then assessed the distribution and the frequency of HLA mutations in a subset of longitudinally genotyped patients relapsing after allo-HCT (N = 48). Specimens at diagnosis (N = 40), at post-chemotherapy (N = 9) and post-allo-HCT (N = 44) relapse were sequenced for both HLA mutations and myeloid genes (Source Data 1, Table 2). We used a newly implemented pipeline designed to recognize somatic mutations and allelic losses within HLA loci[29]. Whereas somatic HLA hits were present in 22% (N = 9/40) of patients at diagnosis and in 38% (N = 17/44) of patients at post allo-HCT relapse, in cases with relapse following chemotherapy no HLA hits were found (Fig. 3a, Source Data 2). In the 2 cases harboring HLA aberrations at diagnosis, no hits were found at relapse after chemotherapy (N = 2; Fig. 3a).

HLA lesions were classified as missense, non-sense, frameshift, splicing or losses and were distributed across class I and II alleles (Source Data 2). Most affected loci were A, C, and DQB1 with allelic losses occurring predominantly in DRB1 and DQB1 (Fig. 3b). Mutations and losses were seen irrespectively of donor type and transplant setting (N = 10/20, 50% in matched related; N = 2/8, 25% in haploidentical;

N = 5/17, 29% in unrelated transplants, Fig. 3c). Of note and in line with previous results, HLA hits occurred in the mismatched haplotype in the haploidentical setting. With a median time to post-transplant disease recurrence of 5.8 months (1.67–56.3), HLA aberrations were more frequent in patients with late (>6 months) relapses (42% vs 29% in early relapses; Fig. 3d, e).

When we analyzed the frequency and distribution of most common somatic myeloid driver hits (for panel description see Supplementary Data 3) in relationship to HLA lesions, no difference in terms of myeloid landscape in samples *prior* to transplant *vs.* post-transplant relapses were noticed. Therefore, HLA aberrations accounted for the predominant genomic lesions in post-transplant relapses. However, reverse analysis indicated that relapses with newly acquired HLA lesions were enriched in *RUNX1, DNMT3A, EZH2, EP300* mutations (Fig. 3g) as compared to those with wild type HLA. Of note is that the acquisition of HLA mutations was not related to HED. In fact, no differences were found with regard to the global, class I, II and locus specific divergence between HLA wild type and mutant sub-cohorts, suggesting that HED was not indicative of the immune pressure or propensity for immune escape in this setting. Similarly, mutated HLA loci did not show higher HED as compared to respective loci in healthy controls (Supplementary Fig. 6).

Notably, all patients sequenced at relapse showed a prevalent recipient chimerism, along with the fact that specific HLA events were not found in recipient prior to transplant. These observations indicated that HLA mutations were somatic and occurred at relapse.

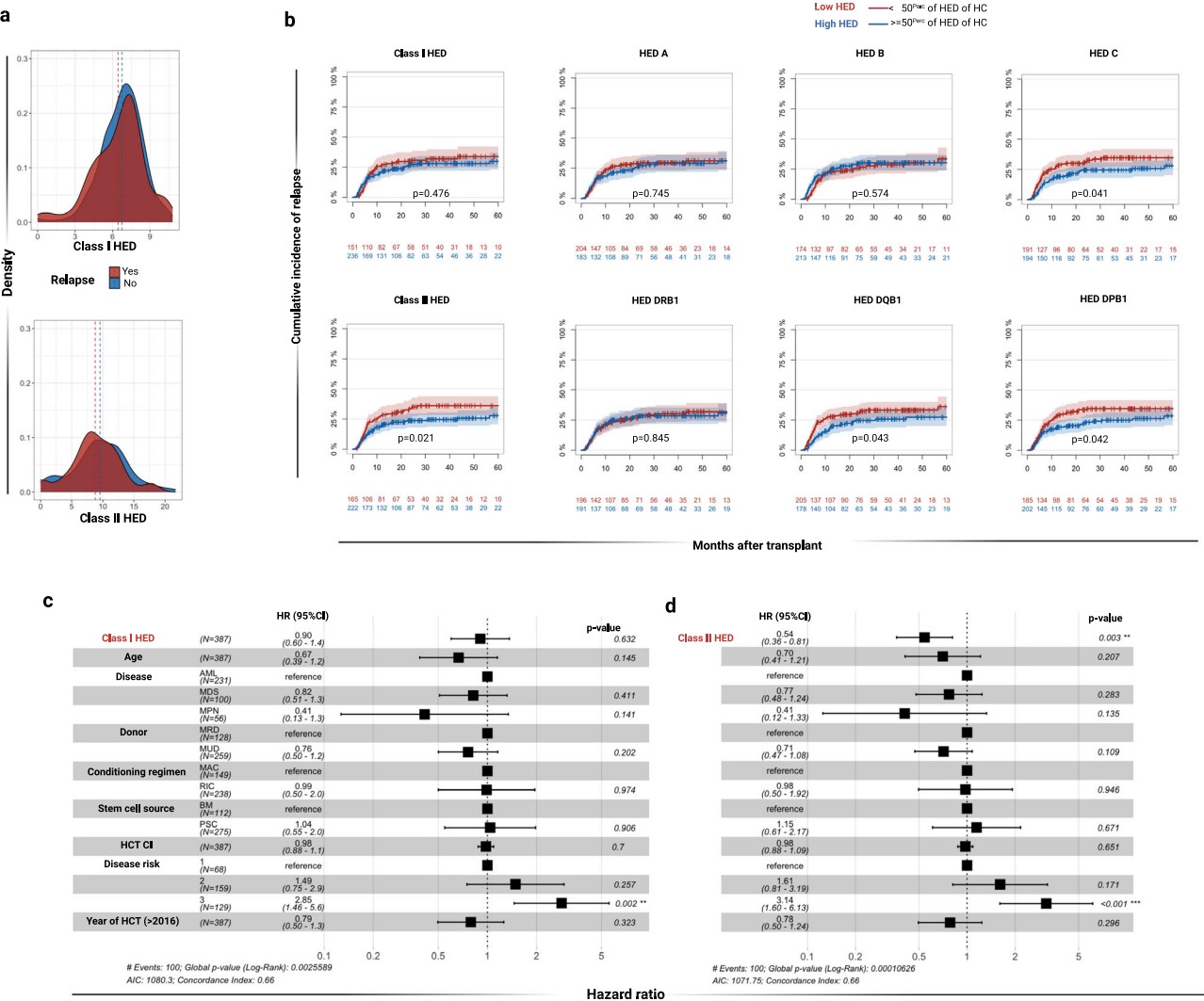

**Fig. 2 | Impact of HED on cumulative incidence of relapse. a** Density distribution of class I and class II HED scores in post-HCT relapsed (red) and non-relapsed patients (blue). **b** Impact of per-class and per-locus HED scores on cumulative incidence (CI) of relapse. Low (red) and high (blue) HED are defined according to the 50th percentile of the corresponding locus in healthy controls. Shaded bands represent 95% confident interval. Non-adjusted *p*-values indicate the significance of the log-rank test. **c** Cause specific multivariable cox regression analysis of relapse including class I HED as main effect variable. **d** Cause specific multivariable cox regression analysis of relapse including class I HED as main effect variable. In (**c**) and (**d**) disease risk was intended as: 1 = low; 2 = intermediate; 3 = high. Black squares indicate the odd ratio and error bars the 95% confident intervals. All the *p*-values were two sided. Source data 1. Abbreviations: HCT-CI hematopoietic cell transplant comorbidity index, AML acute myeloid leukemia, MDS myelodysplastic syndromes, MPN myeloproliferative neoplasms, MRD matched related donor, MUD matched unrelated donor, MAC myeloablative conditioning regimen, RIC reduced intensity conditioning regimen, HR Hazard ratio 95% CI: 95% confident interval.

Donor lymphocyte infusion (DLI) for the treatment of relapse, was performed in half of the patients of this subgroup (18 AML and 7 MDS) at a median time of 7.8 (range 2.9–61.08) months after transplant. DLI were administered mostly in matched related context (14, 56%). A complete response was seen in 5 patients (20%). Nine patients of this subcohort harbored HLA aberrations both in class I and II alleles spanning across exonic, intronic or untranslated (UTR) regions. Of them 6 with exonic mutations or allelic losses did not achieve response to DLI, while 3 patients harboring UTR mutations in HLA-DPA1, -DPB1 and -A were still alive at >1 year post-relapse with controlled disease after DLI and other immunochemotherapy-based strategies (including 5-azacytidine and sorafenib).

**HLA and non-HLA immune transcriptional dysregulation in relapses following allo-HCT**

In addition to somatic HLA hits, we also investigated the transcriptional changes of HLA and immune genes occurring from diagnosis to post allo-HCT relapses in a subgroup of 13 relapsed patients using deep

RNA sequencing of longitudinal (pre and post-transplant) samples (Supplementary Fig. 7; Supplementary Data 4). In post-transplant specimens, HLA genes were often down-regulated, specifically class II loci HLA-DRB1, -DQB1 and -DPB1 (Fig. 4a, Source Data 3). In those with low HLA expression, somatic HLA aberrations were virtually absent, except for 1 case with a mutation in HLA-DRB1 (Fig. 4b). Overall, expression of classical immune checkpoint genes was not significantly altered at relapse, whereas mRNA levels of interferon-gamma response pathway and antigen presentation machinery genes were predominantly decreased (Fig. 4c). In addition, we also observed a down-modulation of Th1 and Th2 cytokine-induced signaling pathways, toll like receptor signaling, neutrophil degranulation, TP53-mediated apoptosis and caspase activation in post-transplant relapses (Supplementary Fig. 8; Supplementary Data 5).

As an illustrative example, we performed single-cell RNA sequencing on pre-sorted CD34+ cells obtained at diagnosis and at post-transplant relapse of an AML patient with *IDH2*, *DNMT3A* and *RUNX1* mutations (Supplementary Fig. 9). CD34 enrichment was

**Table 2 | Patient and transplant characteristics of the geno-typed cohort (N = 48)**

|  |  | N (% or IQR) |
|---|---|---|
| Median Age at HCT |  | 49 (20–73) |
| Disease |  |  |
|  | AML | 35 (73%) |
|  | MDS | 13 (27%) |
| Disease status at transplant |  |  |
|  | CR1 | 32 (67%) |
|  | CR2 | 16 (33%) |
| Donor type |  |  |
|  | MRD | 20 (42%) |
|  | MUD | 20 (42%) |
|  | Haplo | 8 (16%) |
| Graft type |  |  |
|  | BM | 26 (54%) |
|  | PBSC | 22 (46%) |
| Conditioning regimen |  |  |
|  | RIC | 25 (52%) |
|  | MAC | 23 (48%) |
| Median FUP (months) |  | 12.33 (2.4–109.5) |
| Acute GvHD (II-IV) |  | 16 (33%) |
| Chronic GvHD (moderate-severe) |  | 10 (20%) |
| Median time of relapse after HCT (months) |  | 5.8 (1.67–56.3) |
| DLI recipients |  | 25 (52%) |
| Time to DLI (months) |  | 7.8 (2.9–61.08) |
| Extramedullary disease at relapse |  | 4 (8%) |

*IQR* interquartile range, *HCT* hematopoietic cell transplantation, *AML* acute myeloid leukemia, *MDS* myelodysplastic syndrome, *MPN* myeloproliferative neoplasms, *MRD* matched related donor, *MUD* matched unrelated donor, *MMUD* mismatched unrelated donor, *Haplo* haploidentical donor, *BM* bone marrow, *PBSC* peripheral blood stem cells, *MAC* myeloablative, *RIC* reduced intensity, *GvHD* graft versus host disease, *DLI* donor lymphocyte infusion.

performed to mitigate donor cell contamination in this specimen, collected when donor chimerism was <30%. Although no changes in HLA expression and immune checkpoint regulators were found, the top represented down-regulated signatures included an enrichment of IL-1, IL-2, IL-5, TNF and TGF response pathways, indicating an impaired adaptive immune response.

## Low contribution of non-HLA immunogenomic aberrations in post-transplant leukemia relapses

We then asked whether, as in other malignancies, AML/MDS relapses could be constellated by immunogenomic alterations in genes outside HLA loci. To answer this question, we performed whole exome sequencing (WES) on a subset of patients at diagnosis (N = 9), post-chemotherapy relapse (N = 2) and post-HCT relapse (N = 8). These samples were investigated for the presence of mutations affecting antigen presentation and processing machinery and immune checkpoint regulators on target cells (Supplementary Data 5). Applying stringent criteria for variant selection (see methods) and excluding all possibly germline variants, as well as variants with unknown significance (reported in Supplementary Data 6), we did not find a specific enrichment in pathogenic or likely pathogenic somatic aberrations in post-transplant specimens, supporting the concept that direct immune evasion from HLA-related mechanisms represents the major contributor of GvL escape. Of note is that two of these patients (one at diagnosis and the other at post-transplant relapse) were carriers of HLA aberrations, confirmed in the respective WES specimens.

## Evaluation of contribution of HED and KIR ligands on relapse

HLA-C alleles serve as KIR-ligand and can be classified in C1 (Asp80) and C2 groups (Lys80) according to their KIR specificity[30–32]. HLA-C2 homozygosity with or without donor activating KIR2DS2 genotype (or other KIR genotypes) has been shown to increase the risk of relapse in HLA-matched allo-HCT[30]. Thus, the effect of HED in C locus on propensity for relapse could depend on the presence of homozygous C1/C1 or C2/C2 vs heterozygous C1/C2 genotypes. When we classified all the recipient/donor HLA-C alleles of the matched cohort according to the genotypic groups and investigated the clinical outcomes, we did not observe any impact on OS, acute and chronic GvHD and relapse (Supplementary Fig. 10A–D) in univariable models. Similarly, when C haplotype was tested in a multivariate setting (including also class I or C HED, age, donor and conditioning type), we did not observe any contributing effect on relapse, while HED-C continued to affect the risk of recurrence (Supplementary Fig. 10E, F). To complete these observations, KIR genotypes were characterized for 13 matched related donor (MRD)/recipient pairs and described in Supplementary Data 7.

## Discussion

The structural variability of HLA loci along with the presentation of a diversified spectrum of antigens cooperate with a highly heterogeneous TCR repertoire to maintain immune competence[33]. During tumor immunotherapy with immune checkpoint inhibitors, various modes of immune escape have been described including adaptive and genetically-encoded clonal escape[20,34–36]. Similarly, in immune-mediated AA, somatic LOH, deletions and mutations of HLA loci have been described[12–14]. Such pathophysiologic mechanisms may also be operative as acquired resistance factors in allogeneic anti-leukemia surveillance following allo-HCT, wherein either down-modulation of HLA expression or clonal deletion of disparate alleles have been described at relapse[6–8]. Here, through a combination of NGS-based HLA and myeloid genotyping and immune-transcriptomic analyses, we defined more granular molecular mechanisms culminating in immune-resistance phenotypes.

With a previously validated bioanalytic framework[29,37], we were able to identify somatic single nucleotide mutations and small indels in HLA genes along with specific allelic losses. We show that somatic HLA mutations can constitute a potential escape mechanism in patients with MN relapsing after allo-HCT in both matched related and unrelated settings. Indeed, both mutations and deletions of one of the HLA alleles could be detected in fully matched allo-HCT, whereas it is remarkable that in the mismatched setting HLA mutations affected both matched and mismatched alleles. We therefore conclude that the immune selection process leading to clonal escape targets the allele presenting the most immunodominant antigenic peptide, likely not limited to the mismatched ones[6]. Indeed, HLA LOH events often involve large genomic portions of the short arm of chromosome 6, thereby affecting a whole haplotype and not only the mismatched allele and resulting in a global impairment of antigen presentation[4,38–40]. The finding of HLA mutations in relapsed allo-HCT is further supported by the low response rate to DLI. The analysis of the available cohort showed that patients with HLA mutations (including non-UTR alterations) had decreased likelihood of responsiveness to this salvage strategy.

In contrast to tumor control post-immunotherapy, class II HLA genes more than class I were the major drivers of immunologic escape. This result underscores the role of tumor-surveillance of CD4 T-cell specificities[41–44] and antigen-presenting cells, including leukemia-derived dendritic cells[45–48] in GvL effect. As to the function of HLA-C alleles, whose HED scores also influenced the risk of relapse, one can speculate that their allelic heterogeneity determines the degree of natural killer (NK) alloreactivity in the genetic context of inhibitory and stimulatory KIR genes[49–52]. Indeed, our multivariate analysis showed

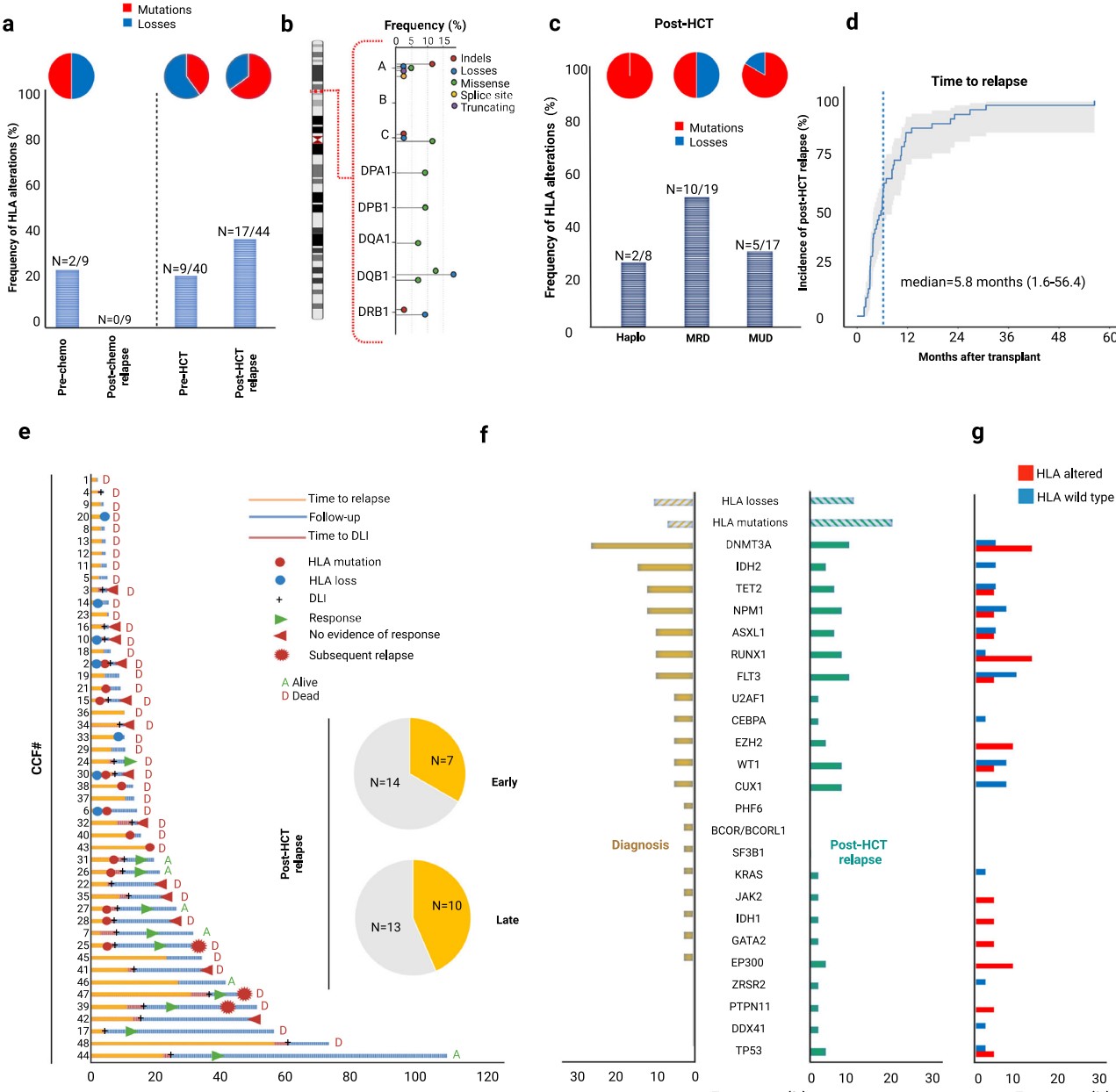

**Fig. 3 | HLA and myeloid landscape of AML and MDS relapsing after allo-HCT. a** Barplot indicating the frequency of HLA mutations in each studied group. Numbers of patients harboring HLA aberrations are reported on the top of each bar. Pie charts capture the frequency of mutations and losses in each group. **b** Lollipop chart showing the frequency of each HLA aberration. The height of the line denotes the frequency, while the color of each dot indicates the type of aberration. **c** Frequency of HLA alterations according to donor group (Haplo: haploidentical donor); MRD (matched related donor); MUD (matched unrelated donor). Pie charts represent the distribution of mutations or losses in each group. **d** Time dependent curve indicating the incidence of relapse of the sequenced group (median time to relapse was 5.8 months). **e** Swimming plot capturing main clinico-biological information for each patient of the sequenced cohort. The yellow barplot indicates the frequency of HLA alterations in late (after 6 months) vs early (before 6 months) post-HCT relapses. **f** Barplots showing the frequency and distribution of HLA mutations, losses and mutations in main myeloid driver genes in AML/MDS at diagnosis (yellow) and at post-HCT relapses (green). **g** Frequency of mutations in myeloid driver genes in HLA-altered vs HLA wild type post-HCT specimens. Source Data 1 and 2.

that locus C HED could influence, over the KIR ligand stratification, the probability of disease recurrence.

A reduced immunopeptidome diversity due to low amino acid divergence at the antigen-binding site level between HLA alleles may impair the GvL effect, configuring a pathophysiological condition similar to the loss of heterozygosity. Indeed, the simple divergence of recipient germline HLA polymorphisms impacts on the risk of relapse. However, high divergence did not translate into selection pressure for the emergence of HLA escape mutants, thus generating an immunogenetic environment able to prevent relapse without favoring HLA

aberrations. To that end, the impact of high HED is also demonstrated by its relationship to the TCR specificity spectrum. For instance, our study also showed that high global mean HED is indeed associated with a broad spectrum of anti-tumor and pathogen-directed TCR specificities. Nevertheless, the locus-specific frequencies of HLA mutations were in line with the findings of the analysis on divergence. Class II loci constantly appeared more mutated than class I genes, with DQB1 and DRB1 alleles strongly compromised by losses and mutations.

Another important aspect of immune-escape relapse after allo-HCT is highlighted by the results of the transcriptional analysis of

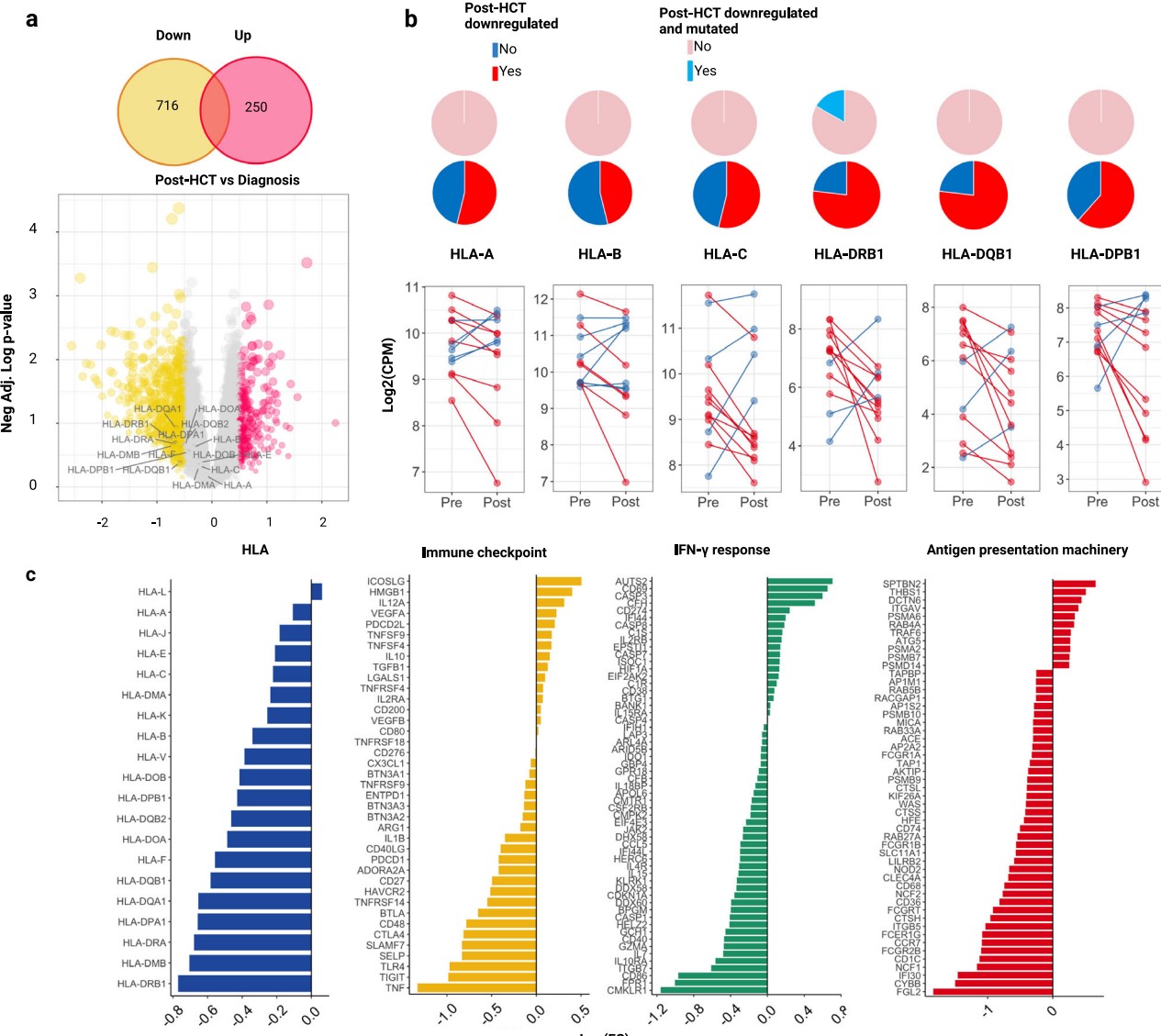

**Fig. 4 | Immune transcriptional dysregulation in post-HCT relapses vs newly diagnosed AML/MDS. a** Results of differential gene expression analysis of bulk RNA samples. Upper panel: Venn diagram showing downregulated (yellow) and upregulated (fucsia) genes. Bottom panel: Volcano plot showing the results of the differential analysis between post-HCT relapses vs newly diagnosed AML/MDS. *X*-axis depicts the logarithmic fold change [log2(FC)] for each gene, while *Y*-axis indicates the negative logarithm of the adjusted *p*-values. The labels highlight HLA genes. **b** Line charts: Paired comparisons of level expression of each HLA gene in pre and post-HCT samples (each paired dot indicates one patient). Patients characterized by a downregulated HLA expression in post-transplant samples are shown in red. Those who did not down-regulate HLA in post-HCT relapses are highlighted in blue. From the bottom to the top, the first series of pie charts highlights the per-locus distribution of downregulated HLA (red) in post-HCT samples. The second series of pie charts shows how many downregulated samples are also affected by HLA mutations/losses (light blue). **c** Barplots indicating the log (FC) of immune genes derived from the differential analysis between post-HCT relapsed vs newly diagnosed AML/MDS (Source Data 3). All the *p*-values were two sided.

paired diagnostic and post-transplant relapse samples. We demonstrate that in some cases, HLA somatic aberrations culminated in lower locus-specific HLA expression, particularly in DRB1. However, downregulation of HLA genes was in most of cases not associated with genomic alterations, underpinning the existence of other non-genomic mechanisms as suggested by previous observations[7,53]. While it is important to point out that baseline HLA expressions may be determined by the genetic subtype of AML (for instance, NPM1 mutant leukemias were reported by us and others to have relatively low HLA-DR expression)[41,54,55], our analysis illustrates the dynamic changes in HLA expression, drastically reduced in most of post-transplant relapses, independently of the molecular disease subgroup.

In a recent study, epigenetic silencing of class II HLA was shown to be regulated by the Polycomb Repressive Complex 2 (PRC2), whose selective inhibition was able to restore HLA class II expression and thus antigen presentation to alloreactive CD4 + T-cells[56]. In addition to HLA down-modulation, our transcriptomic analysis of relapsed leukemia showed an increased spectrum of changes of immunoregulatory and immune response proteins, including those involved in HLA peptide presentation. These changes may be adaptive and potentially reversible, or instead be a result of transcriptional cascades affected by somatic mutations.

In sum, our study unravels the immune escape nature of post-transplant relapse in a significant proportion of patients and provides insights concerning the role in such a setting of the germline and

somatic dysfunction of HLA heterogeneity[33]. In particular, we demonstrate that germline-determined class II HLA divergence and somatic class II HLA mutations, indels or losses can enable an environment of GvL resistance, immune evasion and unfavorable outcomes. Future studies exploring larger cohorts of patients are warranted to better establish how germline and somatic HLA dysfunction may inform daily clinical practice.

## Methods

### Study design and IRB approval

This is a retrospective study aiming at dissecting the genetic determinants of immune escape in AML and MDS patients undergoing allogeneic HCT; three centers contributed to this effort providing clinical and biological data and material: Cleveland Clinic (CCF, Ohio, USA), Nancy Hospital (France), Washington University in St. Louis (WashU, Missouri, USA), (Supplementary appendix, Supplementary Fig. 1; Table 1). We integrated immunogenomic and transcriptomic data from 494 patients (Table 1; Data Source 1) who received allo-HCT for AML ($N = 294$), MDS ($N = 125$) and MPN ($N = 75$) and followed in all three institutions: CCF ($N = 344$), Nancy Hospital ($N = 143$) and WashU ($N = 7$). Pre-transplant clinical HLA genotyping was available in 487 patients and was used as benchmark for HED computation. An ad-hoc targeted high throughput sequencing for classical HLA loci and myeloid genes was performed on selected samples at disease onset ($N = 40$), post-chemotherapy relapse ($N = 9$) and post-transplant relapse ($N = 44$). In addition, 13 patients presenting with disease recurrence were profiled by means of RNA sequencing on sequential diagnosis/post-transplant relapse samples (median 179 days post-HCT, IQR 129–297), whereas single cell transcriptomics was performed on paired specimens for a patient with early relapse (day +92). This research was conducted under the Institutional Review Board (IRB) and local ethics committees of the three centers. All the procedures involving human subjects were carried out under the legacy and the ethical principles of the Declaration of Helsinki.

### Sample collection

Blood or bone marrow (BM) specimens at diagnosis, post-chemotherapy or post-HCT relapses were collected in ethylenediaminetetraacetic acid (EDTA) tubes, and cryopreserved until further use after Ficoll-Paque isolation and suspension in Dimethyl sulfoxide (DMSO) and Fetal bovine serum (FBS) containing media. For each sample we prioritized DNA extraction for HLA genotyping, and targeted myeloid high throughput sequencing. T-cell receptor immunosequencing, was performed on a subset of selected samples, taking into account recipients' HED scores. In case of residual cells, RNA extraction and sequencing were performed. BM specimens collected at the moment of diagnosis and post-transplant relapse for a patient experiencing early relapse were profiled for single-cell RNA sequencing, after flow-based CD34+ cell separation.

### Genotyping studies

HLA sequencing and myeloid targeted sequencing were performed on patient samples included in the biorepository of the Translational Hematology and Oncology Research Department at CCF.

Genomic DNA was isolated directly from cryopreserved unfractionated peripheral or BM blood mononuclear cells with the Nuclei Lysis Solution (Promega) according to manufacturer's instructions.

HLA targeted sequencing was performed with TruSight HLA v2 (Illumina, San Diego California) as indicated previously[29,57]. In brief, 11 HLA loci (Class I HLA-A, B, and C; Class II HLA-DRB1/3/4/5, HLA-DQA1, HLA-DQB1, HLA-DPA1, and HLA-DPB1) were amplified with a long-range polymerase chain reaction (PCR). After amplification, a transposon-based DNA tagmentation was applied to generate DNA amplicons, via DNA fragmentation and addition of adapter sequences. Additional PCR steps provided sequence adapters and indexing

primers to generate sequencing-ready DNA libraries. Prepared libraries were then loaded directly onto a MiSeq System for sequencing. SSO-PCR methods were used to generate HLA genotypes for patients transplanted before 2013. HLA genotyping data used for HED computation were issued from the histocompatibility laboratories of the institutions involved and were coded based on a 2 or 4-field (4–6 digits) nomenclature[58].

Myeloid targeted sequencing was performed on 94 specimens paralleled sequenced for HLA, using a custom panel for detection of myeloid somatic variants from three sequencing platforms: TruSight, TruSeq, and Nextera (Illumina, San Diego, CA, USA). Forty-one most frequent leukemia associated genes common to the three panels were considered for this study (Supplementary Data 3). Sequencing libraries were generated according to an Illumina paired-end library protocol. The enriched targets were sequenced using a HiSeq 2000 or MiSeq (Illumina, San Diego, CA, USA). Paired-end sequenced reads were aligned with the Burrows-Wheeler Aligner (BWA)[59] to GRCh37 reference and post-alignment processing included sorting, marking of duplicates, indexing, base recalibration, according to Genome Analysis Tool Kit v.4 best practices[60]. Variant calling was performed with HaplotypeCaller and VarScan2[60,61]. Variants were annotated using Annovar. Variants with minimum coverage less than 20 or number of high-quality reads less than 5 were filtered out. An in-house developed bio-analytic pipeline identified somatic/germline mutations using sequences derived from controls and mutational databases such as dbSNP138, 1000 Genomes or ESP 6500 database, and Exome Aggregation Consortium (ExAC) as previously published[62–64]. Median coverage of the myeloid panel was 826x. For our variant calling algorithm, the threshold of 20 reads' depth was set-up in previous studies from our group[62,65,66]. Our somatic mutation detection algorithm was previously validated and demonstrated an overall accuracy of 98.7% through comparison with PCR sanger sequencing[65,67].

Whole exome sequencing was applied to a subset of samples studied for the presence of molecular events in other immune non-HLA genes, related to antigen presentation machinery and T cell activation and potentially involved in immune escape (Supplementary appendix).

### HLA mutational analysis

The bioinformatic approach to investigate somatic HLA mutational status is reported in the supplementary appendix and has been previously described[29,57].

### KIR genotyping and HLA-C status assignment

KIR genotyping was performed as previously described, using a sequence-specific oligonucleotidic probe platform (One Lambda, West Hills, CA) and sequence-specific primers (Thermo Fisher Scientific, Waltham, MA)[68,69]. Recipient HLA-C status was assigned according to the allelic C group configurations: I) C1 homozygous: if patients carried only alleles belonging to the supertypes: C*01, C*03, C*07, C*08, C*09, C*10, C*12, C*14, C*16, C*17; C2 homozygous, in case of presence of supertypes C*02, C*04, C*05, C*06, C*15 and C1/C2 heterozygous in presence of a alleles of both groups[30,31].

### TCRβ chain sequencing and analysis

Immunosequencing of the CDR3 regions of human TCRβ chains was performed using the ImmunoSEQ Assay (Adaptive Biotechnologies, Seattle, WA), as previously described[70–72]. In brief, extracted genomic DNA was amplified in a bias-controlled multiplex PCR, with (i) a first PCR step consisting in forward and reverse amplification primers specific for every V and J gene segment, to allow the amplification of the hypervariable CDR3 region, and (ii) a second PCR adding a proprietary barcode sequence and Illumina adapter sequences. CDR3 libraries were sequenced on an Illumina MiSeq system according to the manufacturer's instructions. ImmunoSeq Analizer 3.0 suite was used

for sample export and preliminary statistics and quality control steps while R Bioconductor[73] environment and Immunarch R[74] suite were used for all the downstream analyses as previously described (Supplementary Data 8)[21].

## HED computation

High-quality, protein level (2nd field) HLA data in patient and control cohorts were used as input for HED computation. HED scores were generated for all the subjects and genotypes in the study using the algorithm published by Pierini and Lenz (https://sourceforge.net/projects/granthamdist/) for the calculation of the amino acid sequence divergence at the antigen binding sites of HLA molecules[75]. Briefly, starting from a dictionary including all the protein sequences of exons 2 and 3 for class I alleles and exon 2 for class II alleles, assembled from the IPD-IMGT/HLA database v.3.41 we calculated HED for 6 class I (A, B, C) and II HLA loci (DRB1, DQB1, DPB1). The scores so computed were used for all the downstream analyses.

## Bulk RNA sequencing and analysis

Paired diagnosis/post-transplant relapsed samples obtained from 13 patients (7 previously reported[53]) served for transcriptomic studies. For this last group of samples, total RNA was purified with the NucleoSpin RNA kit (Takara Bio USA, Inc.; Mountain View, CA, USA) according to the manufacturer's instruction. RNA quality check was performed with Agilent 2100 Bioanalyzer. mRNA was enriched using oligo(dT) beads and then was fragmented randomly by adding fragmentation buffer. The cDNA was synthesized using mRNA template and random hexamers primer, after which a custom second-strand synthesis buffer (Illumina, dNTPs, RNase H, and DNA polymerase I) was added to initiate the second-strand synthesis. After a sequential process of terminal repair, a ligation, and sequencing adapter ligation, the double-stranded cDNA library was completed through size selection and PCR enrichment. Illumina technology (NovaSeq 6000) was used for sequencing, after pooling, and >30 million reads were acquired for each sample. FastQC was used to check sequenced reads quality. After trimming and adapter removal, raw reads were mapped to the human hg19 reference genome using RNA STAR[76]. For the transcriptomic study, cohort assembly was performed starting from read counts for all the samples in study. Batch effect was removed through a two-way ANOVA algorithm[77] via limma package prior to further analyses, taking into account the design matrix used to describe comparisons between the samples (diagnosis vs post-HCT relapse), to mitigate the effects derived from different sequencing batches.

Genes that were lowly expressed across >80% of the samples were filtered out. For each sample a normalization factor was calculated through the trimmed mean of M values (TMM) method and final logarithmic counts per million were calculated (log2CPM). Differential expression and gene set enrichment analysis were assessed using edgeR 3.32.1 and limma 3.46 with R computational environment (v. 4)[78]. Benjamini-Hochberg procedure was used for multiple testing correction.

## Single-cell RNA sequencing and analysis

Pre-sorted CD34+ cells BM cells from patient CCF#8 were processed for single-cell library preparation as previously described[79]. Library preparation was performed as per manufacturer instructions with the Illumina Nextera XT DNA sample preparation kit (Illumina, San Diego, California). Briefly a first tagmentation reaction was carried out on 1 ng of cDNA at 55 C for 10 min. The amplification of adapter-ligated fragments was performed using Nextera PCR master mix, index primers and barcodes were added to identify each cell for 15 cycles. The PCR products from each cell were then pooled together and purified using Sera-mag SpeedBeads (0.6:1 ratio). A final quality check of the cDNA library was performed using an Agilent high-sensitivity DNA chip, obtaining a broad peak between 300–800 bp. Single cells were

sequenced using 100 bp paired end sequencing on the HiSeq2500. Raw reads were assessed using FastQC v0.11.5 and aligned to the hg19 human reference genome using the STAR aligner v2.5.3a, after trimming and adapter removal. Rsubread v1.32.4 was used to assemble read matrices from aligned bam files. Cells with fewer than 150,000 reads, along with <65% unique mapping and <35% exonic region coverage, were excluded from further analysis. Genes expressed in less than 10% of cells per patient were removed from downstream analysis. Only those cells that passed quality control were included in the downstream analysis. Seurat R package was used on feature-barcoded matrices for clustering and differential analyses and for visualization purposes.

## Statistical analyses

Median, interquartile ranges (IQR), mean and 95% CI intervals were used where appropriate. Frequency and distribution of categorical variables were expressed as percentage. For all relevant comparisons, after testing for normal distribution, comparative analyses between two groups were performed by Wilcoxon matched-pair signed rank test at 95% CI. Fisher's exact test or Chi-square were applied for independent group comparisons.

Each HED variable was categorized in high and low HED based on the 50th percentile cutoff of the distribution in healthy controls.

Probabilities of survival for OS, defined as the time from transplantation to death for any cause or last follow-up, was calculated using Kaplan-Meier estimates[80], with differences between the curves based on log-rank tests for univariate comparisons. Cumulative incidence of acute and chronic GVHD and relapse were calculated in a competing risk setting, where death was considered the competing event[81].

Multivariate cox regression models were built on cumulative incidence of relapse retaining class I and II HED as main effect variables.

All statistical tests were two-sided, and a $P$-value < 0.05 was considered statistically significant.

All of the analyses and data visualization were performed using the statistical computing environment R (4.0.0 R Core Team, R Foundation for Statistical Computing, Vienna, Austria) and excel Microsoft 365. BioRender.com was used for figures' assembly.

## Reporting summary

Further information on research design is available in the Nature Portfolio Reporting Summary linked to this article.

## Data availability

All the data that support the findings of this study are available within the Article and Supplementary Files.

Genomic data is available through the dbGAP-controlled access database, accession number dbGaP: phs003235.v1. Access can be granted through dbGAP, and contact can be made to Jaroslaw P. Maciejewski (maciejj@ccf.org). There are no restrictions on who will be granted access. Source data are provided with this paper.

## Code availability

The pipeline used for the HLA mutational study has been deposited in the following repository: https://github.com/SMNPAG/HLA-mutations under the https://doi.org/10.5281/zenodo.7651570.

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

## Acknowledgements

This work was supported by US National Institute of Health (NIH) grants R35 HL135795, R01HL123904, R01 380HL118281, R01 HL128425, R01 HL132071, Edward P. Evans Foundation and The Leukemia & Lymphoma Society TRP Award 6645-22 (all to J.P.M.), Aplastic Anemia and MDS International Foundation, Italian Society of Hematology, Fondation ARC pour la Recherche sur le Cancer, MDS Foundation Tito Bastianello Award (to S.P.); VeloSano Pilot Award and Vera and Joseph Dresner Foundation–MDS (to V.V.). C.G. was supported by a grant from the Edward P. Evans Foundation and the American-Italian Cancer Foundation. We greatly thank Lucia D'Aprano, for her IT assistance in the

development of the HLA mutational pipeline. We thank all the reviewers who contributed to improving this work.

## Author contributions

S.P. designed the study, collected, analyzed and interpreted the data, developed the pipeline for the variant calling in HLA region, performed the bioinformatic and statistical analyses, and wrote the manuscript. C.G. performed NGS experiments, clinical and molecular data collection and participated in the analysis interpretation and critical manuscript revision. C.H. developed NovoHLA and the methodology for the calculation of copy number variation. S.He. helped in optimizing the bioanalytical HLA workflow and developing the associated bioinformatic pipeline. S.Ho. collected clinical and genotyping data. A.A., A.D. performed HLA genotyping in Nancy Hospital. M.D.A., M.T., H.E.C., B.K.H., N.S.M., F.F. participated in patient recruitment and management and helped with clinical data collection and interpretation, gave important intellectual inputs and edited the manuscript. V.V. helped in data collection and interpretation, figure conceptualization, gave important intellectual inputs and edited the manuscript. J.P.M. designed and conceptualized the study, supervised genomic experiments, provided funding and resources, interpreted the data analysis and edited the manuscript.

## Competing interests

The authors declare no competing interests.
