## [Peer Review File · Nature Communications]

REVIEWER COMMENTS

Reviewer #1 (Remarks to the Author): clinical expertise in stem cell transplantation

Pagliuca et al. reported HLA evolutionary divergence (HED) predicted the relapse after HLA-matched allogeneic stem cell transplantation (alloSCT) and described the HLA and immune landscape of post-transplant relapse using various genomic assays, HLA somatic mutations, bulk TCR repertoire, bulk or single-cell RNAseq, and WES. The association of HED and relapse risk after alloSCT was previously reported by several institutes, including the author's group. However, there are some discrepancies between the studies on which HED loci are most predictable for relapse (especially for class I HLA). Thus larger cohort study using registry data may be required to determine the impact of HED at different loci on post-transplant relapse adjusted for multiple variables, as proposed in the study in progress at CIBMTR Immunobiology Working Committee Study # IB21-01

<https://www.cibmtr.org/Studies/Observational/StudyLists/Pages/ObservationalStudy.aspx?OSID=a0J0h00000wUwB4EAK>

To better interpret the data and understand the novelty of the findings, I have several major questions summarized below.

Major

1. Stratification of relapse risk analysis based on genetic risk, MRD status, and conditioning regimen.

The risk of post-transplant relapse is known to have a strong correlation with the genetic risk of each primary disease. For example, AML with TP53 or monosomal karyotype has the highest risk of relapse after alloSCT. Measurable residual disease (MRD) is one of the strongest predictive markers for post-transplant relapse. I don't see ELN risk classification for AML or R-IPSS for MDS, or EBMR/CIBMTR disease status in Table 1. The recipients of the reduced-intensity conditioning regimens have a higher risk of relapse in comparison to the ones who received the myeloablative regimens. I am afraid that the authors may underestimate the role of HED if the analysis is not adjusted for the disease risk or conditioning regimen.

2. Possible confounding factor of KIR mismatch for HED score in HLA-C

HLA-C serves as KIR-ligand, and HLA-C2 homozygosity with or without donor activating KIR2DS2 genotype (or other KIR genotype) is known to increase the risk of relapse in HLA-matched alloSCT. I wonder if the significance of HLA-C HED in this cohort may be driven by HLA-C2 homozygosity. If the authors can access the donor KIR genotype, it would be interesting to see how strongly the HED score can predict relapse compared to KIR mismatch.

Reference: Stringaris K et al. Donor KIR Genes 2DL5A, 2DS1 and 3DS1 are associated with a reduced rate of leukemia relapse after HLA-identical sibling stem cell transplantation for acute myeloid leukemia but not other hematologic malignancies. *Biol Blood Marrow Transplant.* 2010 Sep;16(9):1257-64. doi: 10.1016/j.bbmt.2010.03.004. Epub 2010 Mar 17. PMID: 20302958; PMCID: PMC3801172.

Venstrom et al. HLA-C-dependent prevention of leukemia relapse by donor activating KIR2DS1. *N Engl J Med.* 2012 Aug 30;367(9):805-16. doi: 10.1056/NEJMoa1200503. PMID: 22931314; PMCID: PMC3767478.

3. Selected post-transplant samples (n=24) were analyzed for TCR repertoire and clonotype annotation in relation to HED score. I believe this section may represent the novel data in this manuscript. So I would like to know better about the data and analysis.

a. What were the status of GVHD and systemic immunosuppressant use when the samples were collected for TCR repertoire/diversity analysis.

b. Do you have paired pre-transplant donor sample for TCRseq? If so, did you observe any expansion or emergence of certain clonotypes after alloSCT? Especially have you observed the expansion of pathogen-recognizing or cancer-related clonotypes? Any differences between HED high vs low group?

c. Pathogen-recognizing clonotypes can be affected by post-transplant infection or virus reactivation, especially CMV reactivation. Could you at least show no significant differences in the incidences of CMV reactivation between HED high vs low group?

d. Do you have access to minor histocompatibility antigen reactive TCR clonotypes? This

information can help understand whether HED high vs low will shape different sets of alloreactive T cell repertoire in HLA-matched alloSCT.

4. Somatic HLA mutations/aberration in longitudinal samples

- a. The HLA sequencing seems to be performed in bulk non-sorted samples. How could you tell whether HLA mutations were derived from recipient-origin leukemia cells? Could you describe the donor chimerism or VAF of mutated genes by NGS in the sample?
- b. For post-haploSCT relapse sample, please check if the somatic HLA hits occurred in the HLA haplotype not shared between donor and recipient or shared one.

5. Somatic HLA mutations in DLI patients. I think the data is very interesting to see none of the patients with exotic mutations or allelic losses responded to DLI. How many patients were they out of 9 patients with HLA aberration? Any relationship to HED and DLI response? Could you describe more detailed clinical data of DLI cohort (n=25) in supplementary materials? Donor type (HLA matched related, unrelated and haplo), underlying disease, response rate.

6. Bulk RNAseq analysis of post-transplant relapse samples (n=13).

- a. Could you describe the details of the clinical characteristics of the samples used for bulk RNAseq? Type of leukemia, ELN classification or cytogenetic/molecular abnormality, and % of leukemia blasts in the specimen.
- b. Baseline HLA expressions may be determined by the genetic subtype of AML. For example, NPM1 mutant AMLs were reported to have relatively low HLA-DR expression as described by Dufva et al.

References: Dufva O et al. Immunogenomic Landscape of Hematological Malignancies. *Cancer Cell*. 2020 Sep 14;38(3):380-399.e13. doi: 10.1016/j.ccell.2020.06.002. Epub 2020 Jul 9. Erratum in: *Cancer Cell*. 2020 Sep 14;38(3):424-428. PMID: 32649887.

Minor:

1. You may consider citing the article that first reported HLA HED was associated with the post-transplant outcome after alloSCT.

Roerden M, Nelde A, Heitmann JS, Klein R, Rammensee HG, Bethge WA, Walz JS. HLA Evolutionary Divergence as a Prognostic Marker for AML Patients Undergoing Allogeneic Stem Cell Transplantation. *Cancers (Basel)*. 2020 Jul 8;12(7):1835. doi: 10.3390/cancers12071835. PMID: 32650450; PMCID: PMC7408841.

Reviewer #2 (Remarks to the Author): expertise in single cell bioinformatics in leukemia

Pagliuca et al. have used multi-omics data to address a key question in leukaemia biology assessing what confers resistance to the graft versus leukemia effect. By taking an approach applied in solid tumours, they examined the association of HLA evolutionary divergence (HED), a metric reflecting the immunopeptidome diversity, with relapse in allo-HCT recipients and also looked at non-HLA aberrations. They show that somatic HLA mutations constitute the most likely potential escape mechanism.

Although most of the analysis is sound, I have the following questions:

In variant analysis (lines 295-296), what is a 'high quality' read and what variant allele frequency (VAF) threshold did you use to call a variant? What was the average coverage of the myeloid panel and what proportion of variants were lost with depth<20? Same questions go for the HLA mutation screening in lines 308-310. Please explain why different thresholds are used and how they have come up with these cut-off values.

In the bulk RNA sequencing, why have they not used TPM to compare expression of genes across samples? Also, importantly, what did the authors need to run batch correction as in Figure S6 (missing from the Methods)? Weren't all samples (including previously published) run through the same RNAseq pipeline, starting from FASTQ files?

The scRNA analysis is limited to one patient but is still valuable. First of all, how do you interpret

the difference of patterns between the tSNE and UMAP plots and what is the reason to do both when subsequent analysis is based on the UMAP only? More importantly, according to Figure 1 (right panel) and Figure S8 legend, the two samples are called pre/diagnosis and post-transplant samples. If that is the case, then isn't the signal (i.e. downregulated pathways in Figure S8) simply a difference of expression levels between the patient and the donor's transcriptome profile?

Some minor points:

Did the authors look at the association of somatic myeloid drivers and HLA alterations? Is there an enrichment of certain myeloid driver genes in patients with HLA mutations.

It would be interesting to know the length of the HLA region LOH events in patients. Could the authors provide a pile up plot of all the LOH events they have found?

It is not clear in Figure 1 how the Diagnosis, Post-Chemo and Post-HCT samples relate among the 55 patients with somatic analysis. Could they redo this part of the figure so it is clear how many are sequential samples from the same patients? This will enhance Figure 1.

Have you considered multiple testing correction for example in statistical significance testing in Figure 2B and other places?

Reviewer #3 (Remarks to the Author): expertise in HLA sequencing

The manuscript by Pagliuca and colleagues is an extensive analysis of how genetic variation may enable immune escape after allogeneic HCT. There is a lot of data presented between the manuscript and supplemental tables and figures, some of which is not significant and could be refined further.

1. Line 79: The use of the term HLA variability is ambiguous. Please clarify.

2. ST1: 7 of the 494 patients included do not have HLA typing data reported. Why were these included? How was HED calculated?

3. ST1: The following HLA alleles do not exist; this data needs checking: DPB1*0131, B*0901, C*0301, DRB1*3101, DQB1*1001.

Two samples report HLA typing data for HLA-C to the first field only (C*12 and C*07)- how was the ARD/HED determined from this? In ST1, but not in ST2, homozygous HLA loci have the allele name represented twice. This would help to differentiate between loci that are genuinely homozygote from those that have been identified as having some HLA loss.

4. ST1: Much of the HLA typing data is reported in nomenclature that ceased to be used in April 2010. The data reported here needs to be updated to the new nomenclature standards, and to new allele names where necessary. Given the age of these datasets as suggested by this nomenclature, please can the authors confirm how the samples were typed? The manuscript suggests NGS, but this was unlikely pre-2010. If not a sequencing-based methodology, what efforts were made to ensure there were no previously unrecognisable protein variants within the ARD sequences of the individuals tested?

5. The HLA typing data is indicative of an older transplant cohort. Please consider adding era of the transplants to table 1. Was any consideration/adjustment made to the analysis to account for differences in transplant protocols that may have affected relapse probabilities?

6. Line 302 and 326 - The terms 8-digit and 4-digit HLA typing are no longer used. Please correct to the relevant 'field' of typing (i.e. 4-digit typing usually means second-field, i.e. protein level).

7. Some of the references are incomplete (17, 18, 25 etc.) Please update.

8. The multi-part figures were not great quality, making interpretation difficult and in some cases

impossible. F4B - the use of light grey as one of the colours in the pie charts meant that it didn't print.

9. Fig S2: Were all cases in the HLA class II heterozygous cohort censored by ~50 months? The number of individuals left in the OS analyses at time points greater than ~50 months are often too small for meaningful analysis and interpretation. Possibly consider presenting the analysis up to this time point.

10. I couldn't see any reference to Table 2 in the text. Possibly consider merging with table 1 and showing whether there were any statistically significant differences between this subgroup and the cohort overall that may have affected your results.

RESPONSE TO REVIEWERS' COMMENTS

Reviewer #1 (Remarks to the Author): clinical expertise in stem cell transplantation

Pagliuca et al. reported HLA evolutionary divergence (HED) predicted the relapse after HLA-matched allogeneic stem cell transplantation (alloSCT) and described the HLA and immune landscape of post-transplant relapse using various genomic assays, HLA somatic mutations, bulk TCR repertoire, bulk or single-cell RNAseq, and WES. The association of HED and relapse risk after alloSCT was previously reported by several institutes, including the author's group. However, there are some discrepancies between the studies on which HED loci are most predictable for relapse (especially for class I HLA). Thus larger cohort study using registry data may be required to determine the impact of HED at different loci on post-transplant relapse adjusted for multiple variables, as proposed in the study in progress at CIBMTR Immunobiology Working Committee Study # IB21-01

<https://www.cibmtr.org/Studies/Observational/StudyLists/Pages/ObservationalStudy.aspx?OSID=a0J0h00000wUwB4EAK>

We thank the reviewer for these nice comments. We agree on the great potential of registry studies especially in clarifying discrepancies between studies. HED is only one aspect of our study and we believe that identification of somatic HLA mutations is a pivotal finding in this work. That said, we cannot predict the future results of ongoing studies, which may or may not come to fruition. We hope that the current works will not be penalized by potential future publications, which are of unknown outcome. Indeed, the proposed CIBMTR study further substantiates the relevance of the current results.

That said to highlight the need of future studies we added a comment in the main text at line 278-279:

“ Future studies exploring larger cohorts of patients are warranted to better establish how germline and somatic HLA dysfunction may inform daily clinical practice.”

To better interpret the data and understand the novelty of the findings, I have several major questions summarized below.

Major

1. Stratification of relapse risk analysis based on genetic risk, MRD status, and conditioning regimen.

The risk of post-transplant relapse is known to have a strong correlation with the genetic risk of each primary disease. For example, AML with TP53 or monosomal karyotype has the highest risk of relapse after alloSCT. Measurable residual disease (MRD) is one of the strongest predictive markers for post-transplant relapse. I don't see ELN risk classification for AML or R-IPSS for MDS, or EBMR/CIBMTR disease status in Table 1. The recipients of the reduced-intensity conditioning regimens have a higher risk of relapse in comparison to the ones who received the myeloablative regimens. I am afraid that the authors may underestimate the role of HED if the analysis is not adjusted for the disease risk or conditioning regimen.

We thank the reviewer for these insights. We added the required disease risk stratification and HCT-score to Table 1. Multivariate analysis shown in Figure 2 was adjusted for the aforementioned confounding factors and for the intensity of the conditioning regimen. Of note is that class II HED (together with high disease risk scores) continued to show an independent impact on relapse in Cox specific multivariable models.

In the new version of the manuscript (lines 113-116) we modified accordingly the related paragraph and Figure 2 now shows the new multivariate models. Please note that to satisfy reviewer#3 we added also the year of transplant to the new models. Legends to the figures and methods were also updated for the integration of these new variables (see supplementary Appendix).

“Class II, but not class I HED, affected the probability of relapse also when considered in cause-specific multivariable regression models (HR: 0.54, 95%CI: 0.36-0.81, p=0.003), adjusted for age, type of donor, conditioning regimen, disease type, stem cell source, HCT-CI score, disease risk and year of transplant (Figure 2 C,D).”

2. Possible confounding factor of KIR mismatch for HED score in HLA-C

HLA-C serves as KIR-ligand, and HLA-C2 homozygosity with or without donor activating KIR2DS2 genotype (or other KIR genotype) is known to increase the risk of relapse in HLA-matched alloSCT. I wonder if the significance of HLA-C HED in this cohort may be driven by HLA-C2 homozygosity. If the authors can access the donor KIR genotype, it would be interesting to see how strongly the HED score can predict relapse compared to KIR mismatch.

Reference: Stringaris K et al. Donor KIR Genes 2DL5A, 2DS1 and 3DS1 are associated with a reduced rate of leukemia relapse after HLA-identical sibling stem cell transplantation for acute myeloid leukemia but not other hematologic malignancies. *Biol Blood Marrow Transplant.* 2010 Sep;16(9):1257-64. doi: 10.1016/j.bbmt.2010.03.004. Epub 2010 Mar 17. PMID: 20302958; PMCID: PMC3801172.

Venstrom et al. HLA-C-dependent prevention of leukemia relapse by donor activating KIR2DS1. *N Engl J Med.* 2012 Aug 30;367(9):805-16. doi: 10.1056/NEJMoa1200503. PMID: 22931314; PMCID: PMC3767478.

This is an outstanding remark that we are pleased to address. We first categorized all HLA-C alleles into the two KIR ligand groups C1 and C2 and we stratified the impact on outcomes according to the KIR ligand recipient status (homozygous C1/C1, heterozygous C1/C2, homozygous C2/C2). As the reviewer is possibly aware, our group invested a lot of efforts to study both KIRs and their ligands and we also refer to our previous studies on the topic (*please see references: PMID: 17310134, 18195688, 34380091*).

We thus evaluated in univariate fashion the impact of this variable on our matched cohort (MRD+MUD) on OS, cumulative incidence (CI) of relapse, acute grade II-IV GvHD and chronic extensive GvHD but we did not observe any differences. Moreover, we built a Cox-cause specific multivariate model to investigate the contribution of KIR ligand status to relapse where class I and only HED of locus C were considered as main effect variables. Neither in the model for class I nor for HED C, KIR ligand status appeared to influence relapse outcome in this cohort of matched related and unrelated transplants. HED of C locus remained in this model significant after adjustment for the other variables.

This analysis is reported in the following figure (S10).

KIR genotypes were available for 13 donor/recipient pairs (13 MRD) of our cohort. Acknowledging that this small sample size precludes an in-deep analysis of the impact of KIR status, we decided to report our results in a descriptive fashion. Relapses were recorded in 5/13. Among them 4/5 exhibited a C1/C2 haplotype and a negative donor KIR2DS1/ KIR2DS2 genotype. We present a Table (S9) with these 13 patients and for each we show 16 KIR genotypes, however we were not able to evaluate with this data the impact of recipient/donor HED C on KIR status for lack of statistical power. This additional analysis is reported in the new version of the manuscript as follows (line 202-214):

EVALUATION OF CONTRIBUTION OF HED AND KIR LIGANDS ON RELAPSE

HLA-C alleles serve as KIR-ligand and can be classified in C1 (Asp80) and C2 groups (Lys80) according to their KIR specificity. ^{1,2} HLA-C2 homozygosity with or without donor activating KIR2DS2 genotype (or other KIR genotypes) has been shown to increase the risk of relapse in HLA-matched allo-HCT. ¹ Thus, the effect of HED in C locus on propensity for relapse could depend on the presence of homozygous C1/C1 or C2/C2 vs heterozygous C1/C2 genotypes. When we classified all the recipient/donor HLA-C alleles of the matched cohort according to the genotypic groups and investigated the clinical outcomes, we did not observe any impact on OS, acute and chronic GvHD and relapse (**Figure S10A-D**) in univariable models. Similarly, when C haplotype was tested in a multivariate setting (including also class I or C HED, age, donor and conditioning type), we did not observe any contributing effect on relapse, while HED-C continued to affect the risk of recurrence (**Figure S10E-F**). To complete these observations, KIR genotypes were characterized for 13 MRD donor/recipient pairs and described in **Table S9**.

Furthermore, we added the following paragraph to the method section (339-345).

KIR GENOTYPING AND HLA-C STATUS ASSIGNMENT

*KIR genotyping was performed using a sequence-specific oligonucleotidic probe platform (One Lambda, West Hills, CA) and sequence-specific primers (Thermo Fisher Scientific, Waltham, MA). ^{3,4} Recipient HLA-C status was assigned according to the allelic C group configurations: I) C1 homozygous: if patients carried only alleles belonging to the supertypes: C*01, C*03, C*07, C*08, C*09, C*10, C*12, C*14, C*16, C*17; C2 homozygous, in case of presence of supertypes C*02, C*04, C*05, C*06, C*15 and C1/C2 heterozygous in presence of a alleles of both groups. ^{1,2}*

And the following sentence to the discussion (lines 244-245)

“Indeed, our multivariate analysis showed that locus C HED could influence, over the KIR ligand stratification, the probability of disease recurrence.”

3. Selected post-transplant samples (n=24) were analyzed for TCR repertoire and clonotype annotation in relation to HED score. I believe this section may represent the novel data in this manuscript. So I would like to know better about the data and analysis.

a. What were the status of GVHD and systemic immunosuppressant use when the samples were collected for TCR repertoire/diversity analysis.

We thank the reviewer #1 for rising this point. Indeed, to address this issue, we added a new supplemental Table (new Table S3), which gives an overview of GvHD episodes and immune suppression at day 100 and 180 for patients sequenced for TCR repertoire. In this table, we also included CMV risk category and infection (see the comment c below).

b. Do you have paired pre-transplant donor sample for TCRseq? If so, did you observe any expansion or emergence of certain clonotypes after alloSCT? Especially have you observed the expansion of pathogen-recognizing or cancer-related clonotypes? Any differences between HED high vs low group?

We thank the reviewer for this comment. Indeed, some of this data were available in the literature through our previous work (PMID 34236054), where we tracked the divergence of post-transplant TCR repertoires from donor and recipient pre-transplant samples (please see Figure 4 of the aforementioned publication). Nevertheless, to satisfy the reviewer's request we reported now analysis of 3 informative patients for whom now a complete serial pre/donor and post-transplant sampling was available. In the figure below, we specifically tracked the top 10 clones that were overlapping among samples for each patient. Some of these clonotypes were pathogen- or cancer-related, but most of them had an unknown specificity. However, reliable association between HED and expansion of shared clones could not be established.

This figure has been added to the supplementary appendix (Figure S5). We updated the result section as follows (lines 128-132 of the new version of the manuscript):

*“We asked whether the differences in repertoire expansion could be related to a higher frequency of CMV reactivation in relation to HED values. However, HED configuration (high vs low) did not impact on CMV reactivation rates ($p=0.07$ for class I and $p=0.691$ for class II, **Table S3**). Longitudinal clonotype tracking analysis showed that some of these clones were present in pre-transplant or donor samples predating post-transplant hyper-expansion (**Figure S5**).”*

c. Pathogen-specific clonotypes can be affected by post-transplant infection or virus reactivation, especially CMV reactivation. Could you at least show no significant differences in the incidences of CMV reactivation between HED high vs low group?

We absolutely agree with the reviewer about this notion. Indeed, there were no significant differences in terms of class I and II HED in patients sequenced for TCR and reactivating CMV (N=7).

These data are reported in the new supplemental table 3 and a brief comment on this observation has been added to the first paragraph of the results (lines 128-130, see above).

d. Do you have access to minor histocompatibility antigen reactive TCR clonotypes? This information can help understand whether HED high vs low will shape different sets of alloreactive T cell repertoire in HLA-matched alloSCT.

Certainly, it would be very interesting to investigate this aspect in the context of this manuscript. However, at this junction, we were only able to cross-reference IEDB database to determine whether epitopes identified as mHA could be recognized by some known alpha/beta CDR3 sequences. Indeed, for most of the known mHA we were unable to find a specific TCR clonotype. Nonetheless, for one epitope (VLHDDLLEA, encoded by mHA1 gene) we identified several VB CDR3 sequences that were thus searched in our samples: 4 of these clonotypes were expanded in our post-transplant samples. The following table illustrates all the epitopes searched, and the list of TCRs anti HMHA1 identified in our cohort.

MiHA ID	MiHA peptide	Restricted HLA	Chromosome	Gene	Ensembl Gene ID
HA-1/A2	VL[H/R]DDLLEA	A*02:01	chr19	HMHA1	ENSG00000180448
HA-2	YIGEVLVS[V/M]	A*02:01	chr7	MYO1G	ENSG00000136286
HA-8	[R/P]TLDKVLEV	A*02:01	chr9	KIAA0020	ENSG00000080608
HA-3	V[T/M]EPGTAQY	A*01:01	chr15	AKAP13	ENSG00000170776
C19ORF48	CIPPD[S/T]LLFPA	A*02:01	chr19	C19ORF48	ENSG00000167747
LB-ADIR-1F	SVAPALAL[F/S]PA	A*02:01	chr1	TOR3A	ENSG00000186283
LB-HIVEP1-1S	SLPKH[S/N]VTI	A*02:01	chr6	HIVEP1	ENSG00000095951
LB-NISCH-1A	ALAPAP[A/V]EV	A*02:01	chr3	NISCH	ENSG00000010322
LB-SSR1-1S	[S/L]LAVAQDLT	A*02:01	chr6	SSR1	ENSG00000124783
LB-WNK1-1I	RTLSPE[I/M]ITV	A*02:01	chr12	WNK1	ENSG00000060237
T4A	GLYTYWSAG[A/E]	A*02:01	chr3	TRIM42	ENSG00000155890
UTA2-1	QL[L/P]NSVLTLL	A*02:01	chr12	KIAA1551	ENSG00000174718
PANE1	RVWDLPGVLK	A*03:01	chr22	CENPM	ENSG00000100162
SP110	SLP[R/G]GTSTPK	A*03:01	chr2	SP110	ENSG00000135899
ACC-1C	DYLQ[Y/C]VLQI	A*24:02	chr15	BCL2A1	ENSG00000140379
ACC-1Y	DYLQ[Y/C]VLQI	A*24:02	chr15	BCL2A1	ENSG00000140379
P2RX7	WFHHC[H/R]PKY	A*29:02	chr12	P2RX7	ENSG00000089041
ACC-4	ATLPLLCA[R/G]	A*31:01	chr15	CTSH	ENSG00000103811
ACC-5	WATLPLLCA[R/G]	A*33:03	chr15	CTSH	ENSG00000103811
LB-APOBEC3B-1K	[K/E]PQYHAEMCF	B*07:02	chr22	APOBEC3B	ENSG00000179750
LB-ARHGDI1-1R	LPRACW[R/P]EA	B*07:02	chr12	ARHGDI1	ENSG00000111348
LB-BCAT2-1R	QP[R/T]RALLFVIL	B*07:02	chr19	BCAT2	ENSG00000105552
LB-EBI3-1I	RPRARYY[I/V]QV	B*07:02	chr19	EBI3	ENSG00000105246

LB-ECGF-1H	RP[H/R]AIRRPLAL	B*07:02	chr22	TYMP	ENSG00000025708
LB-ERAP1-1R	HPRQEIQIALLA	B*07:02	chr5	ERAP1	ENSG00000164307
LB-FUCA2-1V	RLRQ[V/M]GSWL	B*07:02	chr6	FUCA2	ENSG00000001036
LB-GEMIN4-1V	FPALRFVE[V/E]	B*07:02	chr17	GEMIN4	ENSG00000179409
LB-PDCD11-1F	GPDSSKT[F/L]LCL	B*07:02	chr10	PDCD11	ENSG00000148843
LB-TEP1-1S	APDGAKVA[S/P]L	B*07:02	chr14	TEP1	ENSG00000129566
LRH-1	TPNQRQNV	B*07:02	chr17	P2X5	ENSG00000083454
ZAPHIR	IPRDSWWVEL	B*07:02	chr19	ZNF419	ENSG00000105136
HEATR1	ISKERA[E/G]AL	B*08:01	chr1	HEATR1	ENSG00000119285
HA-1/B60	KECVL[H/R]DDL	B*40:01	chr19	HMHA1	ENSG00000180448
LB-SON-1R	SETKQ[R/C]TVL	B*40:01	chr21	SON	ENSG00000159140
LB-SWAP70-1Q	MEQLE[Q/E]LEL	B*40:01	chr11	SWAP70	ENSG00000133789
LB-TRIP10-1EPC	G[E/G][P/S]QDL[C/G]TL	B*40:01	chr19	TRIP10	ENSG00000125733
SLC1A5	AE[A/P]TANGGLAL	B*40:02	chr19	SLC1A5	ENSG00000105281
ACC-2	KEFED[D/G]IINW	B*44:03	chr15	BCL2A1	ENSG00000140379
ACC-6	MEIFIEVFSHF	B*44:03	chr18	HMSD	ENSG00000221887
HB-1H	EKRGSL[H/Y]VW	B*44:03	chr5	HMHB1	ENSG00000158497
HB-1Y	EKRGSL[H/Y]VW	B*44:03	chr5	HMHB1	ENSG00000158497
DPH1	S[V/L]LPEVDVW	B*57:01	chr17	DPH1	ENSG00000108963
UTDP4-1	R[I/N]LAHFFCGW	DPB1*04	chr9	ZDHHC12	ENSG00000160446
CD19	WEGEPPC[L/V]P	DQB1*02:01	chr16	CD19	ENSG00000177455
LB-PI4K2B-1S	SRSS[S/P]AELDRSR	DQB1*06:03	chr4	PI4K2B	ENSG00000038210
LB-MTHFD1-1Q	SSIID[Q/R]IALKL	DRB1*03:01	chr14	MTHFD1	ENSG00000100714
LB-LY75-1K	LGITYR[N/K]KSLMWF	DRB1*13:01	chr2	LY75	ENSG00000054219
SLC19A1	[R/H]LVCYLCFY	DRB1*15:01	chr21	SLC19A1	ENSG00000173638
LB-PTK2B-1T	VYMND[T/K]SPLTPEK	DRB3*01:01	chr8	PTK2B	ENSG00000120899
LB-MR1-1R	YFRLGVSDPI[R/H]G	DRB3*02:02	chr1	MR1	ENSG00000153029

Below please find the list of the 5 CDR3 beta sequences found in our samples and reactive against the first mHA peptide of the list: VLHDDLLEA.

CASLLAGSYNEQFF
CASLLAGGYNEQFF
CASLSSYNEQFF
CASLVRNEKLF
CASSPINEQFF

We prefer not to include these data into our manuscript because we do not think that these limited findings could enhance our understanding of the impact of HLA heterogeneity dysfunction on the risk of relapse.

4. Somatic HLA mutations/aberration in longitudinal samples

a. The HLA sequencing seems to be performed in bulk non-sorted samples. How could you tell whether HLA mutations were derived from recipient-origin leukemia cells? Could you describe the donor chimerism or VAF of mutated genes by NGS in the sample?

At the time of relapse all patients showed a prevalent recipient chimerism. Also, some HLA mutational events (see table S4 for VAF) have not been found in recipient prior to transplant. These observations indicate that HLA mutations were likely somatic and occurred at the time of post-HCT relapse. We have added the following comment (in addition to the modified Table S4; formerly S3), lines 161-163:

“Notably, all patients sequenced at relapse showed a prevalent recipient chimerism, along with the fact that specific HLA events were not found in recipient prior to transplant. These observations indicated that HLA mutations were somatic and occurred at relapse”.

b. For post-haploSCT relapse sample, please check if the somatic HLA hits occurred in the HLA haplotype not shared between donor and recipient or shared one.

Yes indeed, in the two haploidentical HCT cases, class I somatic mutations occurred in the mismatched haplotype. We modified lines 145-148 of the new version of the manuscript as follows:

*“Mutations and losses were seen irrespectively of donor type and transplant setting (N=10/20, 50% in matched related; N=2/8, 25% in haploidentical; N=5/17, 29% in unrelated transplants, **Figure 3C**). Of note and in line with previous results, HLA hits occurred in the mismatched haplotype in the haploidentical setting.”*

5. Somatic HLA mutations in DLI patients. I think the data is very interesting to see none of the patients with exotic mutations or allelic losses responded to DLI. How many patients were they out of 9 patients with HLA aberration? Any relationship to HED and DLI response? Could you describe more detailed clinical data of DLI cohort (n=25) in supplementary materials? Donor type (HLA matched related, unrelated and haplo), underlying disease, response rate.

Another great comment! We added a paragraph in the result section, highlighting the characteristics of DLI recipients (lines 164-171 of the new version of the manuscript):

“Donor lymphocyte infusion (DLI) for the treatment of relapse, was performed in half of the patients of this subgroup (18 AML and 7 MDS) at a median time of 7.8 (range 2.9-61.08) months after transplant. DLI were administered mostly in matched related context (14, 56%). A complete response was seen in 5 patients (20%). Nine patients of this subcohort harbored HLA aberrations both in class I and II alleles spanning across exonic, intronic or UTR regions. Of them, 6 with exonic mutations or allelic losses did not achieve response to DLI, while 3 patients harboring UTR mutations in HLA-DPA1, -DPB1 and -A were still alive at >1 year post-relapse with controlled disease after DLI and other immunochemotherapy-based strategies (including 5-azacytidine and sorafenib).”

6. Bulk RNAseq analysis of post-transplant relapse samples (n=13).

- a. Could you describe the details of the clinical characteristics of the samples used for bulk RNAseq? Type of leukemia, ELN classification or cytogenetic/molecular abnormality, and % of leukemia blasts in the specimen.

A table with the clinical details of the cohort sequenced for RNAseq analysis has been added to the supplementary material (Table S7).

- b. Baseline HLA expressions may be determined by the genetic subtype of AML. For example, NPM1 mutant AMLs were reported to have relatively low HLA-DR expression as described by Dufva et al.

References: Dufva O et al. Immunogenomic Landscape of Hematological Malignancies. *Cancer Cell*. 2020 Sep 14;38(3):380-399.e13. doi: 10.1016/j.ccell.2020.06.002. Epub 2020 Jul 9. Erratum in: *Cancer Cell*. 2020 Sep 14;38(3):424-428. PMID: 32649887.

We absolutely agree with the reviewer on this point. Indeed, in an independent work recently published by our group (PMID: 36499220), we reported a similar behavior for *NPM1*-mutated AML. However, what we wanted to point out with this analysis was the dynamics of changes in HLA expression, that was drastically reduced in most of post-transplant samples, independently of the molecular disease subgroup.

We added a comment to that to the discussion section (lines 262-265):

*“While it is important to point out that baseline HLA expressions may be determined by the genetic subtype of AML (for instance, *NPM1* mutant leukemias were reported by us and others to have relatively low HLA-DR expression),^{5,6,7} our analysis illustrates the dynamic changes in HLA expression, drastically reduced in most of post-transplant relapses, independently of the molecular disease subgroup.”*

Minor:

1. You may consider citing the article that first reported HLA HED was associated with the post-transplant outcome after alloSCT.

Roerden M, Nelde A, Heitmann JS, Klein R, Rammensee HG, Bethge WA, Walz JS. HLA Evolutionary Divergence as a Prognostic Marker for AML Patients Undergoing Allogeneic Stem Cell Transplantation. *Cancers (Basel)*. 2020 Jul 8;12(7):1835. doi: 10.3390/cancers12071835. PMID: 32650450; PMCID: PMC7408841.

We thank the reviewer for this suggestion, we added this and 2 other references of recent work on the role of HED in transplant.

Reviewer #2 (Remarks to the Author): expertise in single cell bioinformatics in leukemia

Pagliuca et al. have used multi-omics data to address a key question in leukaemia biology assessing what confers resistance to the graft versus leukemia effect. By taking an approach applied in solid tumours, they examined the association of HLA evolutionary divergence (HED), a metric reflecting the immunopeptidome diversity, with relapse in allo-HCT recipients and also looked at non-HLA aberrations. They show that somatic HLA mutations constitute the most likely potential escape mechanism.

Although most of the analysis is sound, I have the following questions:

In variant analysis (lines 295-296), what is a ‘high quality’ read and what variant allele frequency (VAF) threshold did you use to call a variant? What was the average coverage of the myeloid panel and what proportion of variants were lost with depth<20? Same questions go for the HLA mutation screening in lines 308-310. Please explain why different thresholds are used and how they have come up with these cut-off values.

We thank very much the reviewer for highlighting this important issue.

VAF threshold for HLA variant calling was 2%. We estimated that under this threshold no reliable variant could be identified in hyper-polymorphic regions such as HLA loci. Coverage of the myeloid panel was 826x. Average coverage for HLA targeted panel was 834x. For our variant calling algorithm, the threshold of 20 reads’ depth was set-up in previous studies from our group.⁸⁻¹¹ Our somatic mutation detection algorithm was previously validated and demonstrated an overall accuracy of 98.7% through comparison with PCR sanger sequencing.^{9,8} We added this comment and relative references to the method section (lines 331-

334). Please note that some paragraphs of this section were moved to the supplementary appendix (see below).

Lines 316-318

Prepared libraries were then loaded directly onto a MiSeq System for sequencing. SSO-PCR methods were used to generate HLA genotypes for patients transplanted before 2013.

Lines 331-334

Coverage of the myeloid panel was 826x. For our variant calling algorithm, the threshold of 20 reads' depth was set-up in previous studies from our group^{9,11} Our somatic mutation detection algorithm was previously validated and demonstrated an overall accuracy of 98.7% through comparison with PCR sanger sequencing.^{9,8}

In the bulk RNA sequencing, why have they not used TPM to compare expression of genes across samples? Also, importantly, what did the authors need to run batch correction as in Figure S6 (missing from the Methods)? Weren't all samples (including previously published) run through the same RNAseq pipeline, starting from FASTQ files?

We thank again the reviewer for these outstanding technical points. As to the choice of normalization method, we preferred CPM to be consistent with the previous publication by Christopher et al, which represents the source for 7 paired samples of our cohort. For the batch correction issue, we estimated that differences between batches (and cohorts) could represent an important bias for our analysis if not adjusted. We therefore applied a two-way ANOVA algorithm included in limma R package to perform correction for this confounding factor.

Cohort assembly was performed starting from read raw counts.

We clarified all these aspects into the RNAseq method paragraph.

Line 364-365: Paired diagnosis/post-transplant relapsed samples obtained from 13 patients (7 previously reported¹²) served for transcriptomic studies.

Lines 375-378: For the transcriptomic study, cohort assembly was performed starting from read counts for all the samples in study. Batch effect was removed through a two-way ANOVA algorithm¹³ via limma package prior to further analyses, taking into account the design matrix used to describe comparisons between the samples (diagnosis vs post-HCT relapse), to mitigate the effects derived from different sequencing batches.

The scRNA analysis is limited to one patient but is still valuable. First of all, how do you interpret the difference of patterns between the tSNE and UMAP plots and what is the reason to do both when subsequent analysis is based on the UMAP only?

tSNE and UMAP differences illustrate the same clusters with two different analytic methods. To avoid confusion, we removed the tSNE panel from the new version of the Figure S9 (Figure S8 in the previous version) in the supplementary Appendix. Figure legend has been modified accordingly.

More importantly, according to Figure 1 (right panel) and Figure S8 legend, the two samples are called pre/diagnosis and post-transplant samples. If that is the case, then isn't the signal (i.e. downregulated pathways in Figure S8) simply a difference of expression levels between the patient and the donor's transcriptome profile?

This is an insightful observation! Indeed, the post-transplant sample undergoing scRNAseq has been collected at the moment of relapse when donor's chimerism was <30% and in presence of a leukemic bone marrow infiltration. Moreover, both samples were presorted to make sure that the hematopoietic compartment studied was mostly represented by blasts. For this reason, we estimate that donor cells in this sample are underrepresented and the analyzed elements constitute a good representation of recipient's leukemia relapsing cells. We clarify this aspect in the respective paragraph of the results:

Lines 183-186 of the new version of the manuscript: “As an illustrative example, we performed single-cell RNA sequencing on pre-sorted CD34+ cells obtained at diagnosis and at post-transplant relapse of an AML patient with IDH2, DNMT3A and RUNX1 mutations (Figure S8). CD34 enrichment was performed to mitigate donor cell contamination in this specimen, collected when donor’s chimerism was <30%.”

Some minor points:

Did the authors look at the association of somatic myeloid drivers and HLA alterations? Is there an enrichment of certain myeloid driver genes in patients with HLA mutations.

Yes indeed, we looked at these associations (panel 3G). The number of events in both groups is not sufficient to reach statistical significance, however we observed a slight enrichment in *RUNX1*, *DNMT3A*, *EZH2*, *EP300* mutations in HLA altered samples. This finding is reported at lines 153-155.

It would be interesting to know the length of the HLA region LOH events in patients. Could the authors provide a pile up plot of all the LOH events they have found?

We thank very much the Reviewer for this comment. As requested, we included a figure showing the deletions spanning HLA locus. Of note is that this figure is derived from aberrations of the coverage ratio per deep NGS of the corresponding loci. We added comments on this matter in supplementary Appendix and a new Figure S11. Corresponding figure legend is included.

The bioinformatic approach for the detection of HLA aberrations appears now in the supplementary appendix and provides deeper explanations of the computation of HLA loss events.

HLA MUTATIONAL ANALYSIS

Details of the bioinformatic approach to investigate somatic HLA mutational status have been described elsewhere.¹⁴ In brief, after obtaining a confident full 4th field typing through NovoHLA typing algorithm (Novocraft Technologies), paired-end reads from either targeting sequencing were directly aligned on a per-patient HLA reference using Novoalign (Novocraft Technologies Sdn Bhd).

This process is performed by allowing i) multi-alignment of high quality reads ii) intermediate options of clipping iii) reasonable mismatch penalties and gap openings facilitating the alignment of reads with a mismatched base or a deletion/insertion. After classical sorting, marking duplicates and indexing procedures, according to Genome Analysis Toolkit V.4 best practices,¹⁵ variant calling was performed using Varscan v2.4 in tumor-only mode.¹⁶ A minimum coverage of 30 reads (>10 reads for the variant allele) was used as a threshold for the detection in the targeted platform while a minimum coverage of 10 reads (>4 for the variant allele) was used for WES samples. This generated a list of variants with false positive events being still potentially present among true calls. To avoid this background noise, we developed a java tool built on the multi-alignment files provided by the HLA-IPD/IMGT database,¹⁷ to retain only non-polymorphic calls (see supplementary appendix). Average coverage for HLA targeted panel was 834x. VAF threshold for HLA variant calling was 2%. We estimated that under this threshold no reliable variant could be identified in hyper-polymorphic regions such as HLA loci.

Allelic loss was imputed computing the number of reads covering each called heterozygous allele within a given locus. The following formula is used:

$$\text{Log}_2 \frac{C_i}{(\sum C_i, C_z)/2}$$

With C_i and C_z representing the read coverage for each allele belonging to the same locus. For structurally similar alleles, we included an adjustment taking into account sequence variation defined as “Variant coverage”, directly computed by the NovoHLA pipeline.

All mean Log2 ratios <-1.5 were retained as confident allelic losses, based on a previous internal validation study on 234 healthy controls and cell lines that did not showed altered regions.¹⁴

It is noteworthy to clarify that since this algorithm is based on the coverage information for each reference base of each allele, differently from SNP-array platforms, does not allow a proper estimation of the genomic boundaries for 6p loss events but can only identify heterozygous allelic losses. A visualization of this process is reported in **Figure S11**.

FIGURE S11: ALLELIC LOSS EVENTS OF PRE AND POST-HCT SPECIMENS. A) Sketch summarizing the all the somatic allelic losses occurring in 6p region in pre and post-HCT samples. **B)** Scatterplot indicating the Logarithmic base 2 ratio of the base coverage for each allele. Data were retrieved from the HLA sequence information of patient CCF#6 at post-HCT relapse. Each dot indicates a reference base coverage for each allele, and the position of each dot on the y-axis represents the log₂ of the base coverage/average locus depth ratio.

It is not clear in Figure 1 how the Diagnosis, Post-Chemo and Post-HCT samples relate among the 55 patients with somatic analysis. Could they redo this part of the figure so it is clear how many are sequential samples from the same patients? This will enhance Figure 1.

We clarified longitudinal samples in the figure as requested.

Have you considered multiple testing correction for example in statistical significance testing in Figure 2B and other places?

All the p-values in Figure 2B represent single log-rank tests for univariate Kaplan Meyer analysis. We considered that for this analysis there was no indication to perform multi-test correction. However, false discovery rate, via Benjamini-Hochberg procedure, has been applied on the output of the RNAseq and differential transcriptomic analyses.

This was better specified at line 382-383.

Reviewer #3 (Remarks to the Author): expertise in HLA sequencing

The manuscript by Pagliuca and colleagues is an extensive analysis of how genetic variation may enable immune escape after allogeneic HCT. There is a lot of data presented between the manuscript and supplemental tables and figures, some of which is not significant and could be refined further.

Commensurate with this and previous reviewers, a lot of these issues have been mitigated.

1. Line 79: The use of the term HLA variability is ambiguous. Please clarify.

We substituted this term with HLA heterogeneity to be also consistent with our previous work (PMID: 36131910).

2. ST1: 7 of the 494 patients included do not have HLA typing data reported. Why were these included? How was HED calculated?

We thank the reviewer for highlighting this issue and we apologize for the lack of clarity. In these 7 patients, HLA typing was not available, but their samples were used for bulk RNAseq analysis. These patients were not included in the HED analysis.

2. ST1: The following HLA alleles do not exist; this data needs checking: DPB1*0131, B*0901, C*0301, DRB1*3101, DQB1*1001.

We apologize for this issue, we corrected the HLA typing in the table.

Two samples report HLA typing data for HLA-C to the first field only (C*12 and C*07)- how was the ARD/HED determined from this? In ST1, but not in ST2, homozygous HLA loci have the allele name represented twice. This would help to differentiate between loci that are genuinely homozygote from those that have been identified as having some HLA loss.

Table S1 describes the HLA genotype of each patient. Alleles reported once in Table S1 were homozygous for that particular locus and all the doubtful homozygous calls were confirmed by repetitive HLA determinations. For the clarity, we corrected this issue in Table S1 that now reports the name of the allele twice for each homozygous call.

4. ST1: Much of the HLA typing data is reported in nomenclature that ceased to be used in April 2010. The data reported here needs to be updated to the new nomenclature standards, and to new allele names where necessary. Given the age of these datasets as suggested by this nomenclature, please can the authors confirm how the samples were typed? The manuscript suggests NGS, but this was unlikely pre-2010. If not a sequencing-based methodology, what efforts were made to ensure there were no previously unrecognisable protein variants within the ARD sequences of the individuals tested?

We thank the reviewer for pointing out this important issue. Indeed, heterogeneity of typing platforms represents a limitation of most registry-based immunogenetic studies (consider for instance work from HCT

consortia such as CIBMTR or EBMT), which is very difficult to mitigate. That said, in our cohort all the patients longitudinally assessed for the mutational status of HLA genes were profiled with an NGS-based HLA sequencing technique, necessary to obtain a high quality 4-field typing. However, the algorithm to compute HED requires a 2-field typing and thus also a clinical-based typing performed prior to 2012 by SSO-PCR was sufficient to compute this metric. Consequently, this strategy was deployed only in about 10% of the whole cohort, since after 2013 NGS was available in both centers involved. It is important to highlight that the protein HLA reference used for the HED computation derived from a recent version of the IMGT database (Version 3.40).

We better clarified this aspect in the method section, and we added the year of transplant and typing (also in response to comment #5).

5. The HLA typing data is indicative of an older transplant cohort. Please consider adding era of the transplants to table 1. Was any consideration/adjustment made to the analysis to account for differences in transplant protocols that may have affected relapse probabilities?

Transplants were performed between 2010 and 2021. We added the year of transplant to Table 1 and we adjusted our multivariate analysis for this variable, without observing any change on the impact of class II HED.

Multivariate models are reported here for the reviewer's convenience and Figure 2 has been updated.

6. Line 302 and 326 - The terms 8-digit and 4-digit HLA typing are no longer used. Please correct to the relevant 'field' of typing (i.e. 4-digit typing usually means second-field, i.e. protein level).

The terms have been updated.

7. Some of the references are incomplete (17, 18, 25 etc.) Please update.

Done

8. The multi-part figures were not great quality, making interpretation difficult and in some cases impossible. F4B - the use of light grey as one of the colours in the pie charts meant that it didn't print.

We apologize for this issue. We updated Figure 4B

9. Fig S2: Were all cases in the HLA class II heterozygous cohort censored by ~50 months? The number of individuals left in the OS analyses at time points greater than ~50 months are often too small for meaningful analysis and interpretation. Possibly consider presenting the analysis up to this time point.

All the panels in Figure S2 were redone based on the X-axis cutoff requested.

10. I couldn't see any reference to Table 2 in the text. Possibly consider merging with table 1 and showing whether there were any statistically significant differences between this subgroup and the cohort overall that may have affected your results.

We thank the reviewer for this suggestion. We have now merged Tables 1 and 2. Table 2 presents clinical characteristics of the subcohort sequenced for HLA and myeloid mutations. This subgroup of patients relapsed after transplant, and HLA-mediated immune escape in disease recurrence represents the main aspect of this part of the manuscript. We considered for this analysis only AML and MDS and we excluded MPN patients to confer clinical homogeneity. There were no differences in terms of graft sources, donor type and conditioning regimens between this group and the whole cohort, however type of disease and disease status differed for the reasons we just mentioned.

REFERENCES

1. Venstrom, J. M. *et al.* HLA-C –Dependent Prevention of Leukemia Relapse by Donor Activating *KIR2DS1*. *N Engl J Med* **367**, 805–816 (2012).
2. Stringaris, K. *et al.* Donor KIR Genes 2DL5A, 2DS1 and 3DS1 Are Associated with a Reduced Rate of Leukemia Relapse After HLA-Identical Sibling Stem Cell Transplantation for Acute Myeloid Leukemia but Not Other Hematologic Malignancies. *Biology of Blood and Marrow Transplantation* **16**, 1257–1264 (2010).
3. Sobecks, R. M. *et al.* Survival of AML patients receiving HLA-matched sibling donor allogeneic bone marrow transplantation correlates with HLA-Cw ligand groups for killer immunoglobulin-like receptors. *Bone Marrow Transplant* **39**, 417–424 (2007).
4. Hong, S. *et al.* Influence of Killer Immunoglobulin-Like Receptors and Somatic Mutations on Transplant Outcomes in Acute Myeloid Leukemia. *Transplantation and Cellular Therapy* **27**, 917.e1-917.e9 (2021).
5. Dufva, O. *et al.* Immunogenomic Landscape of Hematological Malignancies. *Cancer Cell* **38**, 380-399.e13 (2020).
6. Pagliuca, S. *et al.* Comprehensive Transcriptomic Analysis of VISTA in Acute Myeloid Leukemia: Insights into Its Prognostic Value. *Int J Mol Sci* **23**, 14885 (2022).
7. Ferraro, F. *et al.* Immunosuppression and outcomes in adult patients with de novo acute myeloid leukemia with normal karyotypes. *Proc Natl Acad Sci USA* **118**, e2116427118 (2021).
8. Makishima, H. *et al.* Dynamics of clonal evolution in myelodysplastic syndromes. *Nat Genet* **49**, 204–212 (2017).
9. Hirsch, C. M. *et al.* Molecular features of early onset adult myelodysplastic syndrome. *Haematologica* **102**, 1028–1034 (2017).
10. Nagata, Y. *et al.* Invariant patterns of clonal succession determine specific clinical features of myelodysplastic syndromes. *Nat Commun* **10**, 5386 (2019).
11. Nagata, Y. *et al.* Machine learning demonstrates that somatic mutations imprint invariant morphologic features in myelodysplastic syndromes. *Blood* **136**, 2249–2262 (2020).
12. Christopher, M. J. *et al.* Immune Escape of Relapsed AML Cells after Allogeneic Transplantation. *New England Journal of Medicine* **379**, 2330–2341 (2018).
13. Smyth, G. K. & Speed, T. Normalization of cDNA microarray data. *Methods* **31**, 265–273 (2003).
14. Gurnari, C. *et al.* Is nature truly healing itself? Spontaneous remissions in Paroxysmal Nocturnal Hemoglobinuria. *Blood Cancer J* **11**, 187 (2021).
15. Van der Auwera, G. A. *et al.* From FastQ data to high confidence variant calls: the Genome Analysis Toolkit best practices pipeline. *Curr Protoc Bioinformatics* **43**, 11.10.1-11.10.33 (2013).
16. Koboldt, D. C. *et al.* VarScan 2: Somatic mutation and copy number alteration discovery in cancer by exome sequencing. *Genome Res.* **22**, 568–576 (2012).
17. Robinson, J. *et al.* IPD-IMGT/HLA Database. *Nucleic Acids Research* gkz950 (2019) doi:10.1093/nar/gkz950.

REVIEWERS' COMMENTS

Reviewer #1 (Remarks to the Author):

The authors made satisfactory responses to my comments and modified the manuscript accordingly. I thank the author's time and effort in addressing each point with comprehensive supplementary data. The current manuscript now shows a very robust dataset to support author's interpretation of the genomic landscape in patients with post-transplant relapse. This manuscript would be a pivotal paper to provide an excellent resource for future research in post-transplant relapse.

Reviewer #2 (Remarks to the Author):

The authors have addressed all of my concerns and have no further comments.

Reviewer #3 (Remarks to the Author):

The authors have addressed most of my comments. One point that remains is that of HLA nomenclature. While I appreciate the limitations imposed by studies spanning a wide time-frame, it is important for research involving HLA data to adhere to the guidelines and standards set out for this data by the WHO Nomenclature Committee for Factors of the HLA System (<https://hla.alleles.org/nomenclature/index.html>). I would therefore encourage the authors to update their HLA typing data to the current nomenclature.